# Multiverse Predictions for Habitability: Fraction of Life That Develops Intelligence

**McCullen Sandora** [1,2]

[1]  Institute of Cosmology, Department of Physics and Astronomy, Tufts University, Medford, MA 02155, USA; mccullen.sandora@gmail.com
[2]  Center for Particle Cosmology, Department of Physics and Astronomy, University of Pennsylvania, Philadelphia, PA 19104, USA

**Abstract:** Do mass extinctions affect the development of intelligence? If so, we may expect to be in a universe that is exceptionally placid. We consider the effects of impacts, supervolcanoes, global glaciations, and nearby gamma ray bursts, and how their rates depend on fundamental constants. It is interesting that despite the very disparate nature of these processes, each occurs on timescales of 100 Myr-Gyr. We argue that this is due to a selection effect that favors both tranquil locales within our universe, as well as tranquil universes. Taking gamma ray bursts to be the sole driver of mass extinctions is disfavored in multiverse scenarios, as the rate is much lower for different values of the fundamental constants. In contrast, geological causes of extinction are very compatible with the multiverse. Various frameworks for the effects of extinctions are investigated, and the intermediate disturbance hypothesis is found to be most compatible with the multiverse.

**Keywords:** multiverse; habitability; mass extinctions

## 1. Introduction

This is a continuation of the work initiated in [1–3] aimed at advancing progress on the multiverse hypothesis by connecting it to biological and geological notions of habitability to generate predictions which will be testable within the timespan of several decades. Our technique has been to use detailed criteria for what life needs to count the number of environments suitable for life in the universe, and to check how this depends on the fundamental constants of physics. This counting depends on the assumptions we make about what constitutes a habitable environment, and several of the choices we made imply the existence of much more fertile universes, which would host the majority of observers. When the use of a habitability criterion leads to a very small probability of our existence in this universe, we conclude that this criterion is incompatible with the multiverse hypothesis. Thus, the existence of the multiverse can be used to predict which notions of habitability are right or wrong. While there is currently no way to distinguish among competing notions of habitability, our knowledge on this front is advancing rapidly, and future telescopes and space missions are slated to greatly elucidate what conditions are required for life. We will ultimately determine the exact conditions for habitability, and when this occurs, we may check how this compares to the predictions the multiverse has made. Since there are a great number of factors to consider, there are several independent testable predictions that will serve to test this otherwise almost untestable hypothesis.

The main tool for estimating the number of observers within a universe is the Drake equation, which is a product of factors associated with stellar, planetary and biological habitability. As such, our analysis was (relatively) neatly split by this compartmentalization, with each of our previous papers devoted to a separate domain. While the number and properties of stars, and as of recently the planets orbiting them, are quite well known, as we progress through the factors the analysis

becomes more speculative. In this paper, we focus our attention on the fraction of biospheres that develop intelligent societies. Without wading too much into the debate of what exactly constitutes an intelligent observer, we adopt the self-aggrandizing view that humanlike intelligence is somewhat representative. The author adopts the rather common view that language may represent an excellent proxy for general purpose intelligence, but the results of this paper do not depend too much on the details of this assumption.

Determining the fraction of life bearing planets that develop intelligence is a monumental task, and we make no claim to doing it justice in this letter. We can begin to estimate the physical effects that can preclude this event, adopting the viewpoint that $f_{int} = 1 - f_{\dagger}$, where $f_{\dagger}$ is the fraction of biospheres that are so affected by cataclysm that intelligence cannot develop. This treats the emergence of intelligence (noogenesis) as otherwise inevitable, though there are many additional factors that could contribute to this that should eventually be folded into the full analysis. Even in our restricted setting, we cannot tabulate all possible catastrophes and runaway processes that can occur on a planet in order to provide a true estimate of the fraction that survive sufficiently long, but we instead highlight a few for which we are able to readily encapsulate the dependence on physical parameters.

*Mass Extinctions*

The history of life on Earth is punctuated by several episodes of mass extinction, where a large fraction of the species present at the time did not survive. In the 540 Myr since the advent of complex life, there have been five of these absolutely catastrophic episodes, as well as 20–30 distinguishable lesser episodes. Although for the majority of the study of natural history these great extinctions were treated as 'different in size but not kind' that were not in need of explanation, it is by now generally understood that they are a product of catastrophic changes in the Earth's environment [4]. The single piece of evidence that shifted public perception so radically was the discovery of the iridium anomaly at the onset of the Cretaceous extinction, which indicated that it was triggered by a massive impactor [5]. While this has subsequently been thoroughly confirmed by the discovery of the associated impact crater at Chicxulub (see [6] for a review), the causes of the earlier mass extinctions remain under intense debate to this day.

Several salient features of the observed mass extinctions are worth pointing out for our purposes. Firstly, they all manifest differently. The fossil records of each indicate that the characteristics of the species that went extinct, in which order, the abruptness, severity, and recovery pattern, were all markedly different from event to event [4]. This indicates that the change in conditions that ultimately triggered the extinction did not arise from the same underlying cause, but instead, each event represents a unique flavor of catastrophe. The list of possible causes includes glaciation, impact, supervolcano, biological innovation, anoxia, sea level change, gamma ray burst, and climate change. Many of these causes can lead to additional other causes, leading to a domino effect of extinctions. They may not be mutually exclusive, and some of the extinctions could well have been the product of several concurrent factors. Secondly, there is no strong consensus for the causes of the earlier events, so that there is still room for speculation on the ultimate cause(s). In fact, every possible environmental trigger for mass extinctions has, at some point or another, been applied to explain every mass extinction. Nevertheless, the differences now known lead to a better understanding of the probable causes for each individual extinction, which we go through in detail now.

The Ordovician extinction (444 Mya), the first to occur, is the one which most clearly occurred in two separate waves, separated by a period of 0.5 Myr [4]. There is strong evidence that this was accompanied by a global cooling, which may either have occurred through the glaciation of a continent drifting over a pole [7], or a nearby gamma ray burst [8]. The Devonian extinction (360 Mya) shows clear signs of ocean anoxia on the ocean shelf, and the primary culprit is the innovation of land plants, which lead to a drastically increased weathering rate on the continents [9]. The Permian (251 Mya), which was the largest extinction, is noted to coincide with the supervolcano that essentially created modern Siberia [10], as well as the development of the chemical pathways

necessary to decompose organic matter that had accumulated on the sea floor for the eons prior, both of which could have drastically altered the climate [11]. The Triassic extinction (200 Mya) is possibly associated with sea level change from the breakup of Pangaea [12], but the cause of this extinction is particularly uncertain. As mentioned earlier, the Cretaceous extinction (66 Mya) was caused by the Chicxulub impact. The cause of the current mass extinction underway today is the most unambiguously established to be the result of a recent biological innovation within homo sapiens that has led to an unprecedented ability to alter the environment [13]. On average, the interval between mass extinctions is around 90 Myr.

In Section 2 we discuss several different models for the biological effect of mass extinctions and the time needed for the biosphere to recover. In Section 3, we discuss the rate of deadly comet impacts, and how this depends on fundamental parameters. In Section 4 we discuss the geological contributions to extinctions, including glaciations, sea level rise, and volcanoes. In Section 5 we discuss the rate of gamma ray bursts. In Section 6 we combine these rates into estimates for the fraction of biospheres that develop intelligence.

## 2. Rates

### 2.1. Catastrophes

Before we begin estimating the rates of various potential catastrophic processes, we first derive the fraction of biospheres that develop intelligence for a given rate of mass extinctions $\Gamma_\dagger$. The full rate can be found as the sum of all the various contributions as

$$\Gamma_\dagger = \Gamma_{\text{comets}} + \Gamma_{\text{glac}} + \Gamma_{\text{vol}} + \Gamma_{\text{grb}} + \dots \tag{1}$$

where the displayed rates are ones we will discuss in the text, though there may potentially be more.

Let us make a brief comment on the implications of the relative rates of each of these, because they all seem to occur with a frequency more or less on the order of 100 Myr-Gyr. This itself is enough to suggest the presence of some selection effect that greatly favors the total rate to be as small as possible. This is reminiscent of a 'law of the minimum', wherein systems wishing to maximize some function of independent variables should only be willing to tune those variables inasmuch as they affect the outcome [14]: in this setting, it would do no good to expend effort making any of these rates arbitrarily small, when the sum will always be dominated by the largest term. Of course, this does not imply the presence of any agent doing the selecting, or indeed even a multiverse: there is plenty of variability within our universe to find a system that happens to be unusually quiet on several different fronts. The purpose of the present paper is to determine to what extent the multiverse is required, or even capable of, explaining this state of affairs.

Now, let us estimate the probability that a biosphere beset by random extinction events develops to the point of intelligence. This will depend on the assumptions for how evolutionary processes operate, and so we will end up with three separate functional forms to test for compatibility with the multiverse. These can be called the setback model, where extinctions cause a relatively short period of reduced biodiversity, the reset model, where extinctions result in a complete loss of progress toward intelligence, and the intermediate disturbance hypothesis model, where there is an optimum rate of extinction. Throughout most of this work we will favor the first model, saving discussion of the other two for Section 6.

Our first model is the setback model of extinctions: here, mass extinctions cause a dramatic reduction of biodiversity that leaves the planet ecologically impoverished for a period of time. After a few speciation timescales, however, niches become repopulated, ultimately reaching the complexity the system exhibited previously. The net effect in this scenario is that the biosphere spends this amount of time in a recovery phase, after which it proceeds as normal. This describes the fossil record well, with the recovery time set by $t_{\text{rec}} \sim 10$ Myr [4].

In this view, extinctions reduce the total amount of time the biosphere can spend exploring evolutionary strategies that may lead to intelligence. Because in [3] we favored a model where the probability of developing intelligent life is linearly dependent on total time (importantly, weighted by the biosphere size, set by total entropy production rate), the fraction of biospheres that develop intelligence would then be proportional to the average amount of undisturbed time. Here, we can model the total amount of habitable time for a system as its stellar lifetime, $t_{hab} \sim t_\star$ (it will actually only be some fraction of this, but this sets the rough timescale). Then the number of extinction events $n$ will be given by a Poisson distribution $p(n)$ with average $\Gamma_\dagger t_{hab}$. A simple estimate for fraction of biospheres that develop intelligence is $f_{int}^{setback} = \mathbb{E}((1 - n/n_{max})\theta(n_{max} - n))$, where $n_{max} = t_{hab}/t_{rec}$ is the number of extinctions which would correspond to the system being bombarded, on average, more frequently than it can recover. This is a somewhat simplified prescription, since it neglects the case where all hits are concentrated in a small interval, after which the system settles down to let life proceed unhindered; however, this situation will be rare, and it suffices to provide a simple formula encapsulating these effects through which the dependence on physical parameters can be tracked. Then we find

$$f_{int}^{setback} = \frac{(\Gamma_\dagger t_{hab})^{n_{max}+1}}{n_{max}! \, n_{max}} \left( e^{-\Gamma_\dagger t_{hab}} + (n_{max} - \Gamma_\dagger t_{hab}) \, \mathrm{E}_{-n_{max}}(\Gamma_\dagger t_{hab}) \right) \tag{2}$$

where $E_a(b)$ is the exponential integral. This somewhat cumbersome expression can be approximated to essentially indistinguishable precision by

$$f_{int}^{setback} \approx \left(1 - \Gamma_\dagger t_{rec}\right) \theta \left(1 - \Gamma_\dagger t_{rec}\right) \tag{3}$$

where $\theta(x)$ is the Heaviside step function. The only difference from the full expression occurs past $\Gamma_\dagger \sim 1/t_{rec}$, so that this approximation discounts incredibly rare survivors. For instance, if $\Gamma_\dagger t_{rec} = 1.3$, the full expression gives $10^{-11}$ rather than 0. The essential feature here is that if the extinction rate exceeds the recovery rate, intelligence will never develop. Given the simplicity of the last expression and its conveyance of this key property, this will be the form we will use.

Next, we discuss a second model of how extinctions affect biospheres, which can be called the reset model. This takes the view that extinction events set the clock back to zero, so that life must start over from scratch each time. In this model, there is a noogenesis timescale $t_{noo} \sim 100$ Myr that is required to develop intelligence, and any progress toward this outcome is erased in the course of a mass extinction. The probability of a biosphere developing intelligence is then simply equal to the probability that a time interval equal to $t_{noo}$ exists at some point during the planet's history for which no mass extinctions occur. In favor of clarity, we opt for a simplified, but easier estimate of the probability: if we break up the total lifetime into $N = t_{hab}/t_{noo}$ intervals and ask what the probability is that at least one of these is undisturbed, we find

$$f_{int}^{reset} = 1 - \left(1 - e^{-\Gamma_\dagger t_{noo}}\right)^{\frac{t_{hab}}{t_{noo}}} \tag{4}$$

This is approximately a step function enforcing $\Gamma_\dagger < 1/t_{noo}$, but with a somewhat large tail.

A third viewpoint is known as the intermediate disturbance hypothesis: this is the notion that instead of life faring best under the most placid circumstances, there is some value of the extinction rate that maximizes biodiversity [15]. This is a relatively standard idea in ecology, though it only holds for some ecosystems [16]. A method of determining which systems it is applicable to has recently been developed [17], though it is certainly still too premature to settle whether this idea holds for the biosphere as a whole. Adopting this view for the entire biosphere is no doubt fueled by the somewhat narcissistic observation that had the dinosaurs never died out, then we mammals would never have

had a chance to radiate, and intelligent life may never have evolved. Nevertheless, it may have merit, and can readily be included in our analysis. For this model, we have

$$f_{\text{int}}^{\text{IDH}} = 4\,\Gamma_\dagger\,t_{\text{dist}}\,\left(1 - \Gamma_\dagger\,t_{\text{dist}}\right) \tag{5}$$

This fraction is maximized at $\Gamma_\dagger = 1/(2t_{\text{dist}})$.

The above three functions are plotted in Figure 1 as a function of $\Gamma_\dagger$. All of them have been normalized to 1 at their maxima, seemingly implying that if biospheres are left alone they are guaranteed to develop intelligence. However, there are surely other factors that may affect this: several that were discussed in [3] are the amount of time a planet spends in the habitable zone, the total entropy it processes, and planet size, though there are undoubtedly many more. We leave any additional considerations to future work, noting that we expect these effects to be largely treatable in a fashion that factorizes from the effects we deal with here.

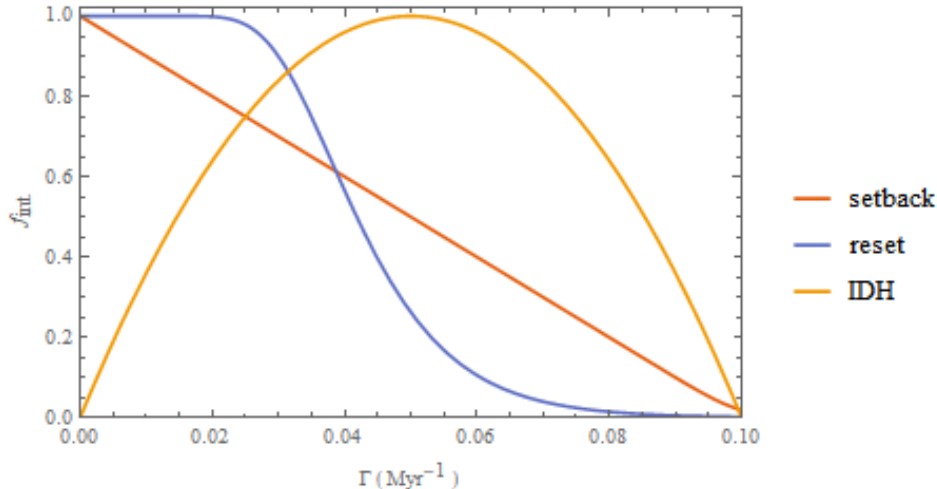

**Figure 1.** Different parameterizations for $f_{\text{int}}$. The red curve is Equation (2), the blue Equation (4), and the orange Equation (5).

## 2.2. What Sets the Recovery Time?

All our analysis relies not only on the extinction rate $\Gamma_\dagger$, but also the recovery time $t_{\text{rec}}$ (or noogenesis time or disturbance time, resp.). Somewhat vaguely, this should be related to the rate of evolution or speciation, but it is far from clear how to connect this to the fundamental properties of physics. Here, we entertain several different educated guesses for what this dependence may be, and then track how the final outcome depends on our choice. The two scenarios are that the recovery time is set by the rate of chemical processes, which dictate the rate of mutations, or else that the recovery time is set by the length of the year, which may dictate the generational timescale for complex organisms.

Our first guess is that the recovery time is set by molecular evolution timescale. This should have some bearing on the rate of genetic mutations, which in turn will dictate how fast speciation can occur. In [18], the molecular timescale was found to be equal to the inverse of the typical molecular binding energy, so if the recovery time is proportional to this, we have

$$t_{\text{rec}}^{\text{mol}} = 3.0 \times 10^{29}\,\frac{m_p^{1/2}}{\alpha^2\,m_e^{3/2}} \tag{6}$$

Here $\alpha$ is the fine structure constant, $m_e$ is the electron mass, $m_p$ is the proton mass, and below $M_{pl}$ is the Planck mass and $\lambda$ is a dimensionless measure of the stellar mass, as in [1]. The large coefficient,

while somewhat off-putting when regarded as a supposed constant independent of physics, is necessary to bridge the gap between these two extremely disparate timescales.

Alternatively, we may take the view that the recovery time is set by the lifespan of macroscopic organisms. It was argued in [19] that this is set by the year length, as organisms would take advantage of nature's cyclic variations for feeding and reproductive purposes. If this is so, then the recovery rate is instead given by

$$t_{\text{rec}}^{\text{year}} = 5.9 \times 10^8 \, \frac{\lambda^{17/8} \, m_p^{7/4} \, M_{pl}^{1/4}}{\alpha^{15/2} \, m_e^3} \tag{7}$$

This again requires a rather large numerical coefficient. These may ultimately be related to the rate of mutations and the size of complex organisms, but we do not go further into detail on these points, because these are more likely dictated by the laws of complex systems rather than the underlying physical substrate.

Though the basis for evolutionary timescales is still not completely known, both of these can be used as hypotheses for our analysis. They lead to different effects: if organism lifespan is truly set by the length of the year, then planets which orbit further from their stars would possess life that lives on a much longer timescale, which would not be observed if it is instead determined by the molecular time. In the other case, however, hotter planets, which have shorter molecular timescales, would have faster evolution. While distinguishing these two scenarios experimentally is probably a ways off (we may like to find a second sample of life first), they do yield differences which are in principle observable.

In the following sections, we will use the setback model and the molecular recovery timescale for definiteness. In Section 6 we will fully explore the alternative choices as well.

## 3. Comets

We now wish to investigate the effect of fundamental parameters on Earth's impact rate. These disruptive events, though rare, can cause a significant effect on complex ecosystems worldwide, as evidenced by their contribution to at least one of the five mass extinctions in Earth's history [4,5]. To get a measure of the rate of impacts, it is essential to know the source of these impactors, as well as their properties.

In general, there are three sources of impacts that may affect our planet. These are asteroids, short period comets, and long period comets. Asteroids are the planetesimal scraps situated between Mars and Jupiter, situated at 2–3 AU. This region of the solar system is in one of Jupiter's orbital resonances, which lead to the conditions that prevent these bodies from condensing into a full planet. Their composition is highly differentiated as a result of primordial heating from a variety of sources, making them rocky. We do not focus on these in this paper, as their properties, such as total number, orbital distribution, and rate of perturbations is highly dependent on details of solar system architecture [20], and so will likely be highly environmentally variable.

There are two populations of comets: those with orbital periods the same order of magnitude as the planets in the outer solar system, and those with orbital periods much larger. Orbits that enter the inner solar system are unstable over geologic time, and so the very existence of such populations indicates a vast reservoir for both. For the short period comets, this is the Kuiper belt, that band of small bodies with orbits on the order of 100 AU that encompasses the dwarf planet Pluto. The long period comets have been inferred to come from an even greater reservoir that extends from 10,000–100,000 AU known as the Oort cloud. This inference is based on both the observed number of long period comets, as well as the orbits of all new comets being practically parabolic rather than hyperbolic, which would be indicative of an extrasolar source (see [21,22] for a reviews). Since these bodies extend so far out into the outer reaches of our solar system, they are routinely influenced by the galactic environment in a variety of ways, which in turn injects them into the inner solar system, with potential to collide with Earth.

The Oort cloud was initially formed out of material from the gas giant region of the solar system. Perturbations from these giant planets caused the originally nearly circular orbits of the comets to elongate secularly over time, increasing the semimajor axis, while preserving the perihelion distance [22]. If this process were allowed to continue indefinitely, most of these bodies would have ultimately been ejected from the solar system into interstellar space. However, once the orbits crossed a threshold of $a_{\text{inner}} \sim 1000$ AU, external perturbations from passing stars also perturb the orbits. These perturbations increase the perihelion out of the inner solar system, thereby preventing any further perturbations from the outer planets occurring. Once this happens, the object is stuck in a nearly stable orbit surrounding our sun until further perturbations either kick it back into the inner solar system, or out of the system completely. Further out, on the order $2-3 \times 10^4$ AU, the inclination or the orbit may also be changed [23]. This leads to a distinction between the inner and outer Oort cloud: inside this, comets orbit within the plane of the solar system, while beyond, the orbits splay into a sphere encompassing our sun.

From observed injection rates, it is estimated that there are approximately $10^{12}$ comets of diameter larger than 1 km within the Oort cloud [22]. At these densities, interactions between Oort cloud objects are negligible for our purposes.

*3.1. Comet Dynamics*

Once a comet is placed in the Oort cloud, its orbit is relatively stable. It is, however, subjected to perturbations that can eventually cause it to reenter the inner solar system, potentially causing an impact on Earth. The galactic tidal force is the dominant cause of reentry, but we first list the other forces for completeness.

Nongravitational forces are mostly caused by the sublimation of ices on the comet, causing the iconic comae that have been observed for thousands of years. This effect only happens once the comet crosses the ice line of the solar system, around 3 AU, and so plays no role while it is in the Oort cloud [21]. Once a comet does reenter the inner solar system, however, these forces play a significant effect on perturbing the orbit. Similarly, planetary perturbations also only play a role once a comet has entered the inner solar system. These were of prime importance for the creation of the Oort cloud [24,25], but not for its subsequent dynamics. The passage of our solar system through molecular clouds has the potential to significantly perturb cometary orbits. However, it was found in [26] that this effect is nonnegligible only for giant clouds, and so can only be expected to be operational once every Gyr or so. Encounters with other star systems can alter the orbits of comets; the density of stars in our galactic neighborhood is 0.185 $M_\odot/\text{pc}^3$, leading to a close encounter of a star within 1 pc of our system about every 100,000 years. The orientation of these encounters is practically random, so that in effect these cause the orbital parameters of the Oort cloud constituents to execute random walks.

The galactic tidal force is caused by the fact that the solar system has finite size, and so the Milky Way exerts a torque throughout the system. In contrast with the effect of close encounters, this effect is directed, causing the orbital parameters to decrease with time (for certain orbits) [22]. The magnitudes of these external influences are proportional to high powers of the semimajor axis $a$, and so these effects are utterly negligible on asteroids and short period comets, but are the most important perturbations on the Oort cloud. These also set the outer boundary of the Oort cloud as the Hill radius of the sun. A rough estimate of this can be found by making the approximation that all the galactic mass is concentrated at a point at the center of the galaxy [27], and is found to be

$$a_{\text{outer}} \sim \left( \frac{M_\odot}{\rho_{\text{gal}}} \right)^{1/3} = 0.0037 \, \frac{\lambda^{1/3} \, M_{pl}}{\kappa \, m_p^2} \tag{8}$$

corresponding to 0.5 pc, or $10^5$ AU[1]. It is possible to do a more sophisticated analysis by modeling the galactic disk: this leads to the conclusion that the Oort cloud is actually an ellipse, but does not change the size of the cloud [23].

The rate of injection into the inner solar system can be computed as [21]:

$$\Gamma_{\text{comets}} = N_{\text{comets}} \, f_{>d_\dagger} \, f_{\text{inj}} \, f_{\text{hit}} \, \frac{1}{P} \tag{9}$$

The quantities in this equation are as follows: $N_{\text{comets}}$ is the total number of comets, $f_{>d_\dagger}$ is the fraction of comets large enough to cause a mass extinction, $f_{\text{tide}}$ is the rate of injection (per orbit, as to be dimensionless), $f_{\text{hit}}$ is the fraction of injected comets that hit the Earth, and $P$ is the period of the comets, given by $P = 2\pi a^{3/2}/\sqrt{GM_\odot}$. In truth these all depend on the radial distribution of comets within the Oort cloud. This is generally expected to decrease with semimajor axis a power law $N(a) \propto a^{-\gamma}$; depending on the processes involved, simulations favor $\gamma = 2.5 - 3.5$ [22] and $\gamma = 3.2 \pm 0.3$ is measured from observations in [28]. We make the simplification, however, that comets orbit at a characteristic radius $a_{\text{Oort}}$. A full treatment would integrate over the orbital radii of Oort cloud objects, weighted by the distribution of semimajor axes- however, the magnitude of the perturbing force depends quite strongly on orbital size, so the integral is dominated by the outermost orbits.

The fraction of comets injected into the inner solar system per period can be computed by considering the secular change in angular momentum, relating this to perihelion distance, and then conditioning on the perihelion to be within the inner system, as in [21]. The radius of the inner solar system is given by the orbits of the outer planets, which is about 15 AU. The giant planets' locations are set as a small multiple of the snow line, $a_{\text{planets}} = 5.6 \, a_{\text{snow}}$, where the snow line was found in [2] to be $a_{\text{snow}} = 30.4 \, \lambda M_{pl}/(\alpha^{5/3} m_e^{5/4} m_p^{3/4})$. However, the injection rate, as computed in [21], is actually independent of this quantity, being given by the ratio of kinetic to gravitational energy $f_{\text{inj}} = \Delta v^2/(2 \, GM_\odot/a_{\text{Oort}})$. This is because the perturbation in orbit is much smaller than the inner system size and is secular, leading to a steady supply of precarious comets right at threshold of entry: the typical change in speed per orbit due to the tidal perturbing force is given by $\Delta v = G\rho_{\text{disk}} aP$. The injection rate is dependent on the square of the perturbing force because for parabolic orbits, angular momentum is proportional to the square root of the perihelion distance $h = (2 \, GM_\odot q)^{1/2}$, and since the distribution of perihelia is uniform, the distribution of angular momentum in linear. When integrating over the entire 'loss cone', the fraction is then proportional to the square of the threshold angular momentum. Combining these elements gives

$$f_{\text{inj}} \sim \left( \frac{\rho_{\text{disk}} \, a_{\text{Oort}}^3}{M_\odot} \right)^2 \tag{10}$$

This expression has the interesting feature that since the Oort cloud radius is set by the typical interplanet spacing in Equation (8), all dependences drop out, and $f_{\text{inj}}$ becomes a pure number.

Once a comet is injected into the inner solar system, more likely than not, perturbations will alter its trajectory to a hyperbolic orbit, ejecting it from the system. The probability that it will impact on any of the rocky planets was found there to be $f_{\text{hit}} = 1.3 \times 10^{-7}$ [21]. This number is set as the ratio of the Earth's gravitational cross section to that of the sun's. Because comets' velocities are orbital, they are larger than the Earth's escape velocity[2] and so the Earth's cross section is given by its geometric value $\sigma_\oplus = \pi R_\oplus^2$. In contrast, the sun's escape velocity is larger than typical comet speeds, so its cross section

---

[1]　Here, and in the following, the expressions from our previous appendices [1,2] are used. Here $\kappa$ is a dimensionless measure of galactic density.

[2]　This hierarchy does not actually hold for all parameters, but the full implications of this will be explored in future work.

is enhanced by gravitational focusing to be $\sigma_\odot = 2\pi G M_\odot R_\odot / v_{comet}^2$. The fraction that impinge on Earth is then given by

$$f_{hit} \sim \frac{R_\oplus^2}{2 R_\odot a_{temp}} = 0.005 \frac{\alpha^6 \beta^{1/2}}{\lambda^{51/20} \gamma^{1/2}} \tag{11}$$

We use here $\beta = m_e / m_p$ and $\gamma = m_p / M_{pl}$.

Now, to estimate $N_{comets}$: the estimated size of the Oort cloud from formation scenarios is $3 - 4\, M_\oplus$, or about $3 \times 10^{11}$ objects of km size or greater [22]. The total mass ejected during planet formation is dictated by the amount of material in the outer regions of the protoplanetary disk, and so is related to the disk density $\Sigma$ by the expression $M_{ejected} \sim 0.01\, \pi a_{planets}^2 \Sigma(a_{planets})$. Most ejected material, however, was removed from the solar system completely. To estimate the amount placed on Oort cloud orbits, [29] found this to be reduced by the factor $a_{planets}/a_{Oort}$. This quantity is dominated by the outermost planets and the innermost Oort cloud orbits, so that Neptune is the most important perturbing agent.

From [30], the inner edge of the Oort cloud can be obtained by setting the timescale of tidal secular evolution equal to the ejection time, and solving for semimajor axis[3]:

$$a_{inner} \sim \frac{M_{Neptune}^{4/3}}{M_\star^{2/3} \rho_{gal}^{2/3} a_{Neptune}} = 1.1 \times 10^5 \frac{\lambda M_{pl}}{\alpha^{5/3} m_e^{5/4} m_p^{3/4}} \tag{13}$$

Here, we have used that the size of ice giant planets is set by the isolation mass evaluated a few times further out than the snow line, such that

$$M_{Neptune} = 3.5 \times 10^{10} \frac{\kappa^{3/2} \lambda^2 M_{pl}^3}{\alpha^{5/2} m_e^{15/8} m_p^{1/8}} \tag{14}$$

Equal to 17 $M_\oplus$.

Then the total Oort cloud mass is

$$M_{Oort} = 1.1 \times 10^{14} \frac{\kappa^2 \lambda^{7/3} m_p^{1/2} M_{pl}^3}{\alpha^{10/3} m_e^{5/2}} \tag{15}$$

To compute the number of comets, we now calculate the typical comet size $d_{comet}$, which is set by the accretion that can occur before ejection. The presence of the ice giants will impart a change in energy to all smaller bodies in their neighborhood; these will then diffuse outward with a timescale of 100 Myr, set by [30]

$$t_{eject} \sim 0.01 \frac{M_\star^2}{M_{Neptune}^2} P = 3.5 \times 10^{-16} \frac{\alpha^{5/2} m_e^{15/8} M_{pl}}{\kappa^3 \lambda m_p^{31/8}} \tag{16}$$

---

3    The inner edge will exceed the outer for stars above the mass

$$\lambda_{none} = 5.7 \times 10^{-12} \frac{\alpha^{5/2} \beta^{15/8}}{\kappa^{3/2}} \tag{12}$$

so that stars larger than this value, corresponding to 17 $M_\odot$ in our universe, will not possess Oort clouds (assuming a similar planetary system architecture to the solar system). However, this bound is of little importance, as it only affects very massive stars unless $\alpha$ or $\beta$ are several times smaller than their observed values.

From [29], the growth rate of a body is given by $\dot{M} = 2\pi G^2 M^2 \rho / c_s^3$. The ambient density can be expressed in terms of the disk surface density as $\rho \sim \Sigma/(c_s a)$, giving the characteristic size of Oort cloud objects to be $M_{\text{comet}} \sim T^2 a/(m_p^2 G^2 \Sigma t_{\text{eject}})$. This leads to a characteristic radius

$$d_{\text{comet}} = 2501 \frac{\kappa^{2/3} \lambda M_{pl}^{5/6}}{\alpha^{25/9} m_e^{3/2} m_p^{1/3}} \tag{17}$$

This is set to be 1 km.

With all these, the rate of cometary impacts on Earth can be estimated. Altogether, this leads to

$$\Gamma_{\text{comets}} = 3.1 \frac{\kappa^{3/2} \alpha^8 m_p^{3/2}}{\lambda^{16/5} m_e^{1/2}} \min \left\{ \left( \frac{d_{\text{comet}}}{d_\dagger} \right)^p, 1 \right\} \tag{18}$$

The normalization here is chosen to reproduce one mass extinction every 90 Myr (which would only be the desirable prescription if comets were the sole cause of extinctions). We have used that the fraction of comets large enough to cause a mass extinction is given by a power law, which holds over 16 orders of magnitude [31]. For the slope we use $p = 1.5$, in agreement with that found in [28], though there is considerable uncertainty in the measurement of this quantity. The only remaining quantity to estimate is the size of comets which will cause mass extinctions.

### 3.2. What Sets the Size of Deadly Comets?

From the geologic record, the size of the smallest impactor that can lead to global disruptions must be slightly more than 10 km, since the Chicxulub impact, at 14 km, caused a mass extinction, whereas the next largest impacts that occurred since the advent of complex life, the Popigai and Manicouagan, both 10 km, did not.

In fact, size is not strictly the sole determiner of the magnitude of the environmental perturbation. Other important factors include the mineral composition of the location of impact: the K-Pg crater location, being a shallow sea, was particularly laden with the mineral gypsum [32], making it an abnormally sulfate-rich site. Additionally, the kinetic energy of the impactor is more relevant, and there is a large spread in the speeds of comets relative to Earth[4]. Nevertheless, in terms of average conditions, the diameter of the impacting comet will dictate the strength of environmental response.

The limiting size depends on the exact mechanism of extinction during an impact event. This is not as straightforward to deduce as one might naively expect, essentially because many Earth systems are catastrophically affected during such a calamity, in many different ways [32]: first, a rather large portion of rock around the initial impact site is vaporized and thrown into the stratosphere. Dust can linger for months, darkening skies to the point where photosynthesis (and even vision) are impossible. Larger rocks can be strewn ballistically over the entire globe, the reentry of which may ignite forests worldwide. Nitrous oxide compounds are created in bulk during initial atmospheric deposition, potentially destroying the ozone layer. Sulfate compounds are liberated during the impact, leading to an intense global cooling that may last decades, and associated acidification that will result in widespread die-offs. If the impact occurs in the ocean, massive tsunamis will wreak havoc on shallow water and coastal ecosystems, and drastically alter the amount of water in the atmosphere. Given this litany of utterly brutal catastrophes, it is small wonder that there are differing ideas to the ultimate cause(s) of extinction.

---

[4]　This is the reason comets can be deadlier than asteroids, even though they are less dense: since they can come from any direction rather than being roughly coorbital with the Earth, the average speed will be $\sqrt{3} v_\oplus = 52$ km/s, rather than $(\sqrt{3} - 2\sqrt{2}) v_\oplus = 12.4$ km/s, a factor of 17.5 more energy. Therefore, comets can be smaller and still impart more energy than asteroids, and so their rate of deadly impacts will be more numerous (which depends on their relative population sizes as well, of course).

The impact that triggered the K-Pg (end Cretaceous) extinction can be of great use here. Though all of these mechanisms are fiercely debated, by now the most plausible seems to be the climatic effect of sulfate injection. It was argued in [33] that not enough submicrometer dust was produced in the impact to cause significant attenuation of sunlight. In [34], it was found that the charcoal record, an indicator of forest fires, was not above the background level at the K-Pg layer, even for sites located in the Americas. It was argued in [35] that marine extinctions are consistent with acidification, rather than darkness-induced productivity collapse. In [36] it was argued that not enough $NO_x$ was created to trigger a significant decrease in ozone layer. Climate modeling in [37] indicates that the amount of $SO_x$ (x = 2,3) created is enough to cause 10–20° global cooling, depending on the uncertain residence time of these molecules. In the following we track the minimum size of an impactor for both the production of $SO_x$ and of dust, especially since the argued relevance of both mechanisms indicates that the threshold diameters may be close in magnitude. Since these scale differently with physical parameters, this coincidence cannot be explained by selection effects operating purely within our universe, and may be a hallmark of anthropic selection.

3.2.1. Sulfate

We begin with the production of sulfate, a gas with the capacity to block sunlight and a residence time of years to decades. To find the minimum size of an impactor necessary for this effect to operate, we relate the optical depth of the sulfate material produced to the mass of the impactor.

Sulfate is generated by the vaporization of the surrounding crater at the impact site. Here, the initial kinetic energy is first converted into breaking molecular binding energy. The number of bonds that can be broken is given by

$$N_{SO_x} \sim \frac{m_i v_i^2}{E_{mol}} \tag{19}$$

Here, $E_{mol}$ is set by the molecular binding energy, though the precise value is determined by assessing exactly how energy is distributed in the surrounding minimum during the initial shock wave [32,38].

Dust and debris created during the impact will rise as a plume high into the atmosphere, nearly vertically. Any that stays within the troposphere will only have a local effect, but the material that makes it to the stratosphere, where horizontal mixing is fast, will cover the Earth within a matter of hours. The energy needed for material to reach the stratosphere is orders of magnitude less than that needed to have a substantial effect on ecosystems, achieved even with paltry atomic bomb blasts [39], so it is a generic feature, of terrestrial planets in any universe, that distribution of material will be global. Here we assume the material will be dispersed relatively uniformly, and so the optical depth will be $\tau = N_{SO_x} \sigma_T / A_\oplus$, where $\sigma_T$ is the molecular cross section. The critical optical depth is uncertain but close to 1, and a combination of this and the energy required per molecule of sulfate can simply be matched with the observed critical comet size in order to find the dependence on physical parameters. We find

$$d_{SO_x} \sim \left( \frac{E_{mol} A_\oplus}{\rho \, v_i^2 \, \sigma_T} \right)^{1/3} = 10.1 \frac{\lambda^{1/4} M_{pl}^{1/2}}{\alpha^3 \, m_e \, m_p^{1/2}} \tag{20}$$

This quantity depends most strongly on the electron mass and the fine structure constant: if either of these were significantly larger, the size of globally catastrophic comets would be much smaller, leading to an enhanced rate. The dependence of this quantity on the physical parameters is plotted in Figure 2, along with the other scales for comparison.

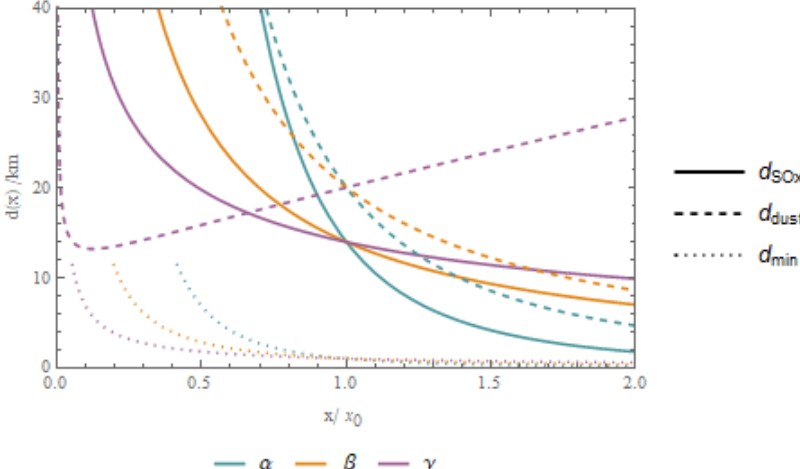

**Figure 2.** The dependence of the various cometary scales on physical parameters. Though these curves scale the same way with $\alpha$ and $\beta$, when varying with respect to $\gamma$ a maximum size is reached close to our observed value. The typical comet size is the smallest of the three except for very small values of the parameters.

### 3.2.2. Dust

Though dust does not appear to have been the primary cause of extinction for the K-Pg event [33], it is still interesting to determine the mass of an impactor that would be required for this mechanism to go into effect, and not just for pure agnosticism toward differing environmental impact models. The reason is that the sensitivity of this hazard to fundamental parameters is much greater than the sulfate case, and so, for different values, dust is the main contribution to extinctions.

The main reason dust is not a major concern for our values is that when the crater is vaporized in the impact event, most of the dust produced is on the order of the grain size of crystals comprising the crust, which is $r_0 = 100\,\mu m$ [33]. Particles of this size have atmospheric residence times on the scale of days, and so any effect will quickly dissipate [32]. Submicrometer dust is required for any lasting impact on the environment, but very little is produced. We first determine how small a dust grain has to be in order to linger in the atmosphere, and then calculate the amount of dust smaller than this that is produced for an impact of a given size.

The residence time of a dust grain can be estimated by dividing the height at which it is deposited in the atmosphere by the rate at which it falls. Most of the plume from the impact is deposited at the base of the stratosphere, and so we simply use a multiple of the atmospheric scale height $H_{atm} = T/(m_a g)$, where $m_a$ is average molecular weight of atmospheric gas. The terminal velocity is given by $v \sim \sqrt{g r_{dust}}$. The residence time is then $t_{res} \sim T/(m_a\, g^{3/2}\, r_{dust}^{1/2})$.

This can be used to determine the largest dust size that is capable of having a lasting impact on the environment, if the threshold residence time is known. However, this immediately becomes a very tricky quantity to define. Even on Earth, a planet we are relatively familiar with, the amount of time each photosynthesizing organism can go without sunlight is highly variable, and how many need to die for a complete ecosystem collapse is a difficult question to address. Nevertheless, several weeks seems to be a reasonable estimate. On other planets, we may assume roughly the same level of hardiness, though this may not be valid if the planet varies drastically, such as on planets which are tidally locked, or have extreme obliquity. On planets in another universe with different fundamental constants, any prediction for how long plants can survive without sunlight should be taken with extreme skepticism. Nevertheless, to make progress, we choose to take the threshold residence time to be several dozen days, where the length of a day is several times larger than the centrifugal limit,

as on Earth. This sets a critical value for the size of dust particles that will affect the atmosphere for a prolonged period as

$$\frac{r_{\text{float}}}{r_0} \sim \frac{T^2}{m_a^2 \, g^3 \, (20 \, t_{\text{day}})^2 \, r_0} = 3.0 \times 10^{-19} \, \frac{\alpha^{1/2} \, \beta^{1/4}}{\gamma} \tag{21}$$

Notice that this is most sensitive to $\gamma$, and if it were much smaller, a significant fraction of the impact would stay in the atmosphere long enough to affect life. The main reason for this is that the pull of gravity will then be weaker, though this effect is partially compensated by the fact that then days will be longer as well. Because of this, smaller comets would be capable of having a drastic effect, and the overall rate would increase.

To relate this to the size of a comet needed, we find the optical depth of the dust injected into the atmosphere, as before. The key difference is that now, of the total amount of dust produced, we are only interested in the amount below this threshold value. The distribution of dust grains is observed to be lognormal distribution over microscopic ranges. Here, in accordance with [32,33,38], we take $r_0 = 100 \, \mu\text{m}$. To set the fraction of submicron dust to be 0.1% from [32], we take $\sigma_{\text{float}} = 1.24$ (whereas if we set the fraction to be 0.01%, as in [33], $\sigma_{\text{float}} = 1.08$). Then the number of particles that remain in the atmosphere is

$$N_{\text{dust}} = \frac{M_{\text{vap}}}{\frac{4\pi}{3} \, \rho \, r_0^3} f_{\text{float}} \tag{22}$$

where $M_{\text{vap}}$ is the total vaporized mass, and the fraction of dust that stays in the atmosphere is

$$f_{\text{float}} = \frac{1}{2} + \frac{1}{2} \text{erfc} \left( \frac{\frac{1}{2}\sigma_{\text{float}}^2 + \log(r_{\text{float}}/r_0)}{\sqrt{2} \, \sigma_{\text{float}}} \right) \tag{23}$$

As with the sulfates above, the optical depth is given by $\tau = N_{\text{dust}} \sigma_{\text{M}}/A_{\oplus}$, except that here the cross section is given by the geometric size of the particles. The threshold values of the optical depth required for photosynthesis and human vision are given in [32] as $\tau = 29$ and $\tau = 143$, respectively. In practice, it matters very little which of these values we take, as the size of comet needed to produce the second effect is only a few times larger than the first. This is

$$d_{\text{dust}} \sim \left( \frac{E_{\text{mol}} \, A_{\oplus} \, r_0}{m_p \, v_i^2 \, f_{\text{float}}} \right)^{1/3} = \frac{1022}{f_{\text{float}} \left( 3 \times 10^{-19} \, \alpha^{1/2} \, \beta^{1/4} \, \gamma^{-1} \right)^{1/3}} \frac{\lambda^{1/4} \, M_{pl}^{1/2}}{\alpha^{5/3} \, m_e \, m_p^{1/2}} \tag{24}$$

For definiteness, this has been normalized to 20 km. This length scale is compared to the size needed for sulfates and the typical comet size in Figure 2.

The size of dangerous comets is the smaller of these two scales, $d_\dagger = \min\{d_{\text{SO}_x}, d_{\text{dust}}\}$. Even though this has a maximum when varying the strength of gravity, when comparing to the dimensionless ratio given by dividing by the comet radius there is no turnover, just a change in slope. More importantly, the quantity $\Gamma_{\text{comet}} t_{\text{rec}}$ does not have a turnover either, as can be seen from Figure 3. There, the probability of observing any value of the constants $\alpha$, $\beta$ and $\gamma$ is displayed, based solely on the number of stars and the fraction of those affected by comets. To give probabilities that are compatible with our observations, this analysis must be included with some of the other factors that affect habitability as discussed in the previous papers of this series. Throughout, we will take the entropy and yellow conditions as our baseline, as detailed in [1,3], and check how extinctions hinder this already viable criterion. A more thorough analysis would run through all the previous combinations to check which are affected by extinctions, but the results of this would be much too cumbersome to report on in this paper.

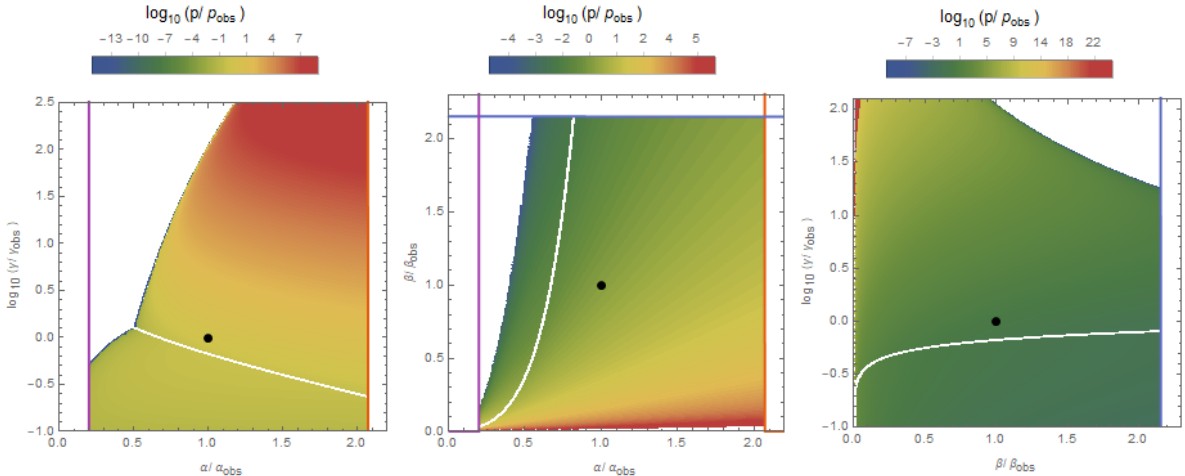

**Figure 3.** Distribution of observers taking comet impact extinctions into account. The black dot denotes values in our universe, and the orange, blue, and purple lines are the hydrogen stability, stellar fusion, and galactic cooling thresholds discussed in [1]. The white line corresponds to the discontinuity where $d_{\text{dust}} = d_{\text{SO}_x}$.

If just the sulfate radius is used, the probabilities of observing our values of the constants are

$$\mathbb{P}(\alpha_{obs}) = 0.07, \quad \mathbb{P}(\beta_{obs}) = 0.17, \quad \mathbb{P}(\gamma_{obs}) = 0.30 \tag{25}$$

where we use the setback model of mass extinctions and the molecular timescale parameterization of recovery time from Section 2. If just the dust radius is used, then

$$\mathbb{P}(\alpha_{obs}) = 0.14, \quad \mathbb{P}(\beta_{obs}) = 0.09, \quad \mathbb{P}(\gamma_{obs}) = 0.28 \tag{26}$$

while for both[5],

$$\mathbb{P}(\alpha_{obs}) = 0.13, \quad \mathbb{P}(\beta_{obs}) = 0.08, \quad \mathbb{P}(\gamma_{obs}) = 0.29 \tag{27}$$

The numbers shift by a factor of two, but there is little difference between these three scenarios. Interestingly, however, even though sulfate production seems to have been the dominant cause of extinction in Earth's previous episodes, the threat of dust more strongly dictates our position in this universe. These should be compared to the values when extinctions are not taken into account, $\mathbb{P}(\alpha_{obs}) = 0.19$, $\mathbb{P}(\beta_{obs}) = 0.44$, $\mathbb{P}(\gamma_{obs}) = 0.32$.

## 4. Volcanism, Glaciations, and Sea Level Change

Three of the other main impetuses of mass extinction, volcanism, glaciations, and sea level change, all depend directly on the amount of heat generated from the mantle. The rate of volcanoes perhaps explicitly, as the yearly output can be directly linked to internal heat, and the observed power law can then be used to extrapolate to the rate of biosphere-altering supervolcanoes. Glaciations and sea level change are set by this tempo as well, as their occurrence is a direct consequence of continental drift leading to a rearrangement of the Earth's land surface capable of triggering a climate instability into a secondary equilibrium. Glaciations as such can occur when an isolated continent is situated over one of the Earth's poles, as occurred with Antarctica 34 Mya, which triggered a glaciation and coincident cooling of the Earth by several degrees [40]. Though the subsequent global extinction was relatively minor, a similar event has been implicated in the Ordovician mass extinction [4], this time

---

5    The code to compute all probabilities discussed in the text is made available at https://github.com/mccsandora/Multiverse-Habitability-Handler.

with the glaciation of Gondwana. Similarly, the original formation of Pangaea lead to a reduction of coastal area, triggering a marine dieoff in the Devonian (and Rodinia as well for stromatolites in the Proterozoic [41]). Since the majority of marine bioproductivity is situated close to continents, any rearrangements have the potential to trigger catastrophic instabilities. Coral reef ecosystems, for example, are extremely sensitive to the changes in available sunlight that accompany sea level change, even by a few meters. Whether by sea level change due to glaciations, or the eventual closing up of intercontinental seaways, both processes depend on the rate of continental rearrangement.

*4.1. Glaciations*

Let us estimate the rate of glaciations/sea level change first. A simple estimate of this is just given by

$$\Gamma_{\text{glac}} \sim \frac{v_{\text{drift}}}{R_{\oplus}} \tag{28}$$

In our previous paper ([3] and references therein), we used an expression for the continental drift rate $v_{\text{drift}} \sim Q/(A_{\oplus}n\Delta T)$, which derives this quantity in terms of the internal heat flow $Q$, number density $n$, area of Earth $A_{\oplus}$, and temperature difference $\Delta T$. This expression then simplifies,

$$\Gamma_{\text{glac}} \sim \frac{Q\,m_p}{M_{\oplus}\,E_{\text{mol}}} = 29.6\,\frac{Q\,m_p^{17/4}}{\alpha^{7/2}\,m_e^{9/4}\,M_{pl}^3} \tag{29}$$

Using $Q = 47$ TW, this indeed yields $\Gamma_{\text{glac}}^{-1} \sim 100$ Myr. Once an expression for $Q$ is used, this can then be incorporated into $f_{\text{int}}$ to determine which values of the constants are compatible with the emergence of intelligent observers.

The Earth's internal heat is somewhat subtle, though, since it depends on time and has multiple distinct sources. If we use the naive dimensional analysis estimate we first found in [3], $Q_{\text{naive}} \sim 92.5\alpha^{9/2}m_e^{7/2}M_{pl}/m_p^{5/2}$, we find that $\Gamma_{\text{glac}}t_{\text{rec}} = 8.3 \times 10^{32}\gamma^2/(\alpha\beta^{1/4})$. If this naive estimate is used, then increasing the strength of gravity by a factor of 3 would increase the rate of glaciations to below the recovery timescale! This is certainly a very big departure from what we've been discussing to this point, where the strength of gravity could vary by two orders of magnitude. Since this was the basis of exclusion for many habitability criteria which favor larger values of this quantity, a drastic reduction in anthropically allowed space such as this would alter our previous conclusions.

However, this sharp boundary is spurious, and disappears if the time dependence of Earth's heat flux is taken into account. For the heat of formation, we found before that

$$Q_{\text{form}} = 2.6\,Q_{\text{naive}}\,\frac{1}{s}\,e^{-s}, \quad s = 10\,\alpha^{1/2}\,\beta^{5/8}\,\gamma\,\sqrt{m_p\,t} \tag{30}$$

When this is used, the exponential dependence on $\gamma$ balances the prefactor, so that stronger gravity no longer obviates the development of complex life.

Above, we only took the heat of formation into account when estimating the rate of glaciations. However, an additional source of heat comes from radioactive elements in the mantle decaying. Indeed, these two sources of heat are comparable [42]. Given this intriguing fact, along with the high sensitivity of this latter form of heat to the fine structure constant uncovered in [43], it is worth thoroughly investigating the interplay between these two quantities for generic values of the parameters.

Generically, the heat generated by radioactivity in the mantle is given by

$$Q_{\text{rad}} = \sum_i f_i\frac{E_i}{t_i}e^{-t/t_i} \tag{31}$$

where $f_i$ is the fraction of species $i$ in the mantle, and the sum is performed over all radioactive species. This is a tall order, especially since the relevant species depend on the constants themselves.

For tractability, we note that in [43], when this sum was performed in earnest, the resultant heat resembled a Gaussian which peaked at $\alpha = 1/144$ at a value 2.7 times our observed heat. This allows us to use a simplified expression

$$Q_{\text{rad}} \sim 5.6 \times 10^{-51} \frac{\alpha^{3/2} \, m_e^{3/4} \, M_{pl}^3}{m_p^{7/4}} f_{\text{rad}}(\alpha), \quad f_{\text{rad}}(\alpha) = 2.7 \, e^{-383.88\left(\alpha - \frac{1}{144}\right)^2} \tag{32}$$

This approximation yields at least 15% accuracy, though much greater for the majority of values[6].

When both sources of heat are used, the probabilities of observing our quantities become

$$\mathbb{P}(\alpha_{obs}) = 0.19, \quad \mathbb{P}(\beta_{obs}) = 0.42, \quad \mathbb{P}(\gamma_{obs}) = 0.30 \tag{33}$$

These are almost indistinguishable from the baseline case, so that including glaciations and other related geological causes of extinctions are fully compatible with the multiverse. Using only one of the sources of heat only alters these numbers by a few percent, whichever we take. The distribution of observers including both sources of heat is displayed in Figure 4.

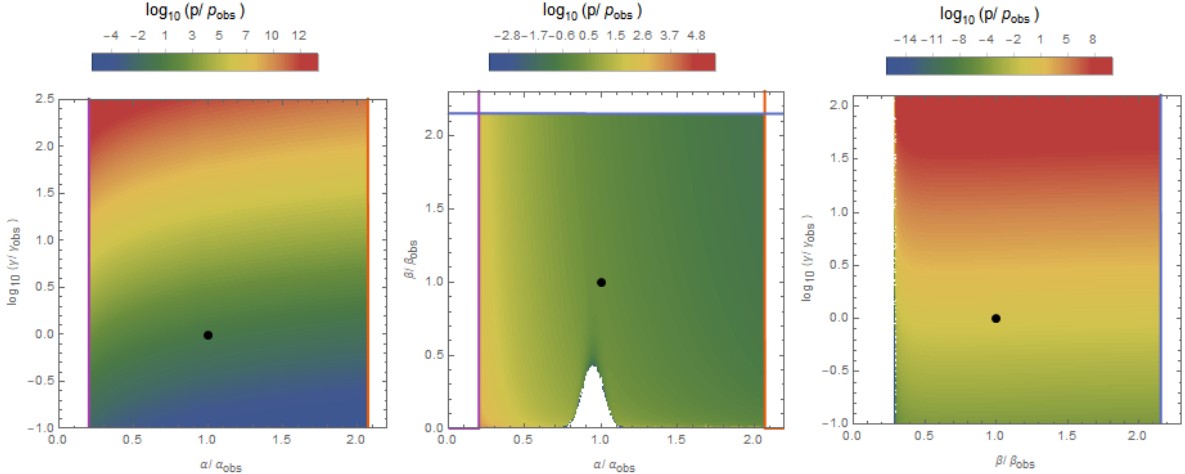

**Figure 4.** Distribution of observers taking glaciations and sea level rise into account. The small white regions for small $\beta$ and $\alpha$ close to our value have such high geologic activity that an extinction would occur every 10 Myr.

*4.2. Volcanoes*

A similar consideration can be made for the rate of supervolcanic eruptions. By these, we mean those rare events that are powerful enough to cause a mass extinction. Volcanic eruptions have been implicated as a causal factor for several of Earth's mass extinctions, most notably the end Permian, which was argued to be a result of the Siberian eruptions, the largest known volcanic event that occurred on land [10].

Volcanoes of all sizes are constantly erupting on Earth, and most are small enough to only affect their immediate vicinity. Some larger ones are capable of exerting a noticeable influence on the whole Earth, but these are correspondingly rare. These large events are qualitatively different in nature from the smaller explosive eruptions, and are most likely a result of a convective instability in the

---

6   This Gaussian approximation can be improved upon by treating the sum as an integral and using the saddle point approximation: this yields $f_{\text{rad}} = 2.7 \, e^{\hat{}}\left(1 + \ell(1-x) - e^{\ell(1-x)}\right)$, with $x = 1/\sqrt{144\alpha}$ and $\ell = 52.72 + \log(t_{\text{rec}}/\hat{t}_{\text{decay}})$, where the hatted quantity inside the log evaluates to 1 for our values. Using this more accurate expression does not affect our results at all.

Earth's mantle [44]. This results in mantle plumes, which are enormous billows originating in the lower mantle, and emplace millions of cubic kilometers of lava on the surface in geologically short periods of time. While these eruptions inject cooling aerosols into the upper atmosphere, these are short lived and are not currently thought to be the major cause of extinction [45]. More likely is the large amount of carbon dioxide, which for the end Permian extinction injected as much as 10 times preindustrial levels, causing an increase in temperature, acidity, and hypoxia [46].

The size and frequency of these events is then dictated by the mantle physics, as outlined in [47]. There, they found the timescale governing both the evolution and periodicity of convective plumes to be

$$t_{\text{conv}} \sim \left( \frac{n \, \nu \, A_\oplus}{g \, \alpha_t \, Q} \right)^{1/2} \tag{34}$$

Here, $\alpha_t \sim 0.02/T_{\text{melt}}$ is the coefficient of thermal expansion, $\nu$ is the viscosity of the mantle, and $g$ is the gravitational acceleration. The only quantity in this expression that requires an explanation of any detail is the viscosity: an expression for this was given in [48] based on the diffusion creep model, where defects in the rock structure such as vacancies migrate in response to stresses. They find

$$\nu = \frac{10 \, T \, d^2}{D_0 \, m_a} \, e^{\Delta H/T} \tag{35}$$

where $d$ is the typical spacing and $D_0$ is the diffusion rate of holes. The exponential term involving the binding energy varies by over three orders of magnitude throughout the mantle, but the overall normalization is dictated by a very large constant offset reflecting the difficulty of deformation. This offset, which is typical of any quantity with Arrhenius type temperature dependence, is difficult to derive without a detailed model of the microscopic physics, but should be relatively independent of the physical constants. Then the final thing to note is that the diffusion constant is given by $D_0 \propto T d^2$, so that $\nu \sim 10^{25}/m_p$. Our final expression for the rate is

$$\Gamma_{\text{vol}} = 3.3 \times 10^{-15} \, \frac{m_p^{15/8} \, Q^{1/2}}{\alpha^{3/4} \, m_e^{3/8} \, M_{pl}^{3/2}} \tag{36}$$

This has been normalized to $1/(90 \text{ Myr})$.

We can also check the typical plume volume, and compare this to that required to have a significant impact on the atmosphere. Again from [47], the characteristic size of a plume is given by

$$\lambda_{\text{plume}} \sim 31 \left( \frac{n \, \kappa_{\text{heat}}^2 \, \nu \, A_\oplus}{g \, \alpha_t \, Q} \right)^{1/4} \tag{37}$$

This corresponds to around 100 km. In [49], it was shown that all plumes are within a narrow range around this value. Here, $\kappa_{\text{heat}}$ is the thermal diffusivity, which was expressed in terms of fundamental constants as $\kappa_{\text{heat}} = 2/(m_e^{1/4} m_p^{3/4})$ in [3]. In terms of constants, the volume of the basalt flow is

$$V_{\text{vol}} = 1.3 \times 10^{23} \, \frac{\alpha^{9/8} \, m_e^{3/16} \, M_{pl}^{9/4}}{m_p^{21/16} \, Q^{3/4}} \tag{38}$$

Most of this will consist of lava which, although devastating for local ecosystems, would not have much impact on the global biosphere. We can estimate how much associated carbon dioxide gas is released by noting that for the end Permian eruption, 2000 km³ basalt flow corresponded to 12 Gt C [50]. The total weight was $10^4$ Gt, so for the total amount of carbon we use $M_C \sim 10^{-3} \rho V_{\text{vol}}$. This can

be compared to the amount needed to significantly warm the climate, which is set by the optical depth becoming appreciable: $M_\text{warm} \sim 44 m_p A_\oplus / \sigma_\text{T}$. Then the ratio of these two masses is given by

$$\frac{M_\text{C}}{M_\text{warm}} = 7.9 \times 10^{19} \frac{\alpha^{57/8} m_e^{43/16} M_{pl}^{1/4}}{m_p^{23/16} Q^{3/4}} \tag{39}$$

If this quantity is less than around 10% of its observed value, then these supervolcanoes will not cause a mass extinction. Though this criteria is somewhat crude, it only affects the overall probabilities by several percent, and so will not be included in our analysis.

If volcanoes are taken to be the only cause of mass extinctions, then the probabilities of observing our values of the constants are

$$\mathbb{P}(\alpha_{obs}) = 0.19, \quad \mathbb{P}(\beta_{obs}) = 0.43, \quad \mathbb{P}(\gamma_{obs}) = 0.30 \tag{40}$$

From here, it can be seen that volcanoes have very little impact on these values, so that this mechanism for mass extinctions is compatible with the multiverse. The distribution of observers is shown in Figure 5.

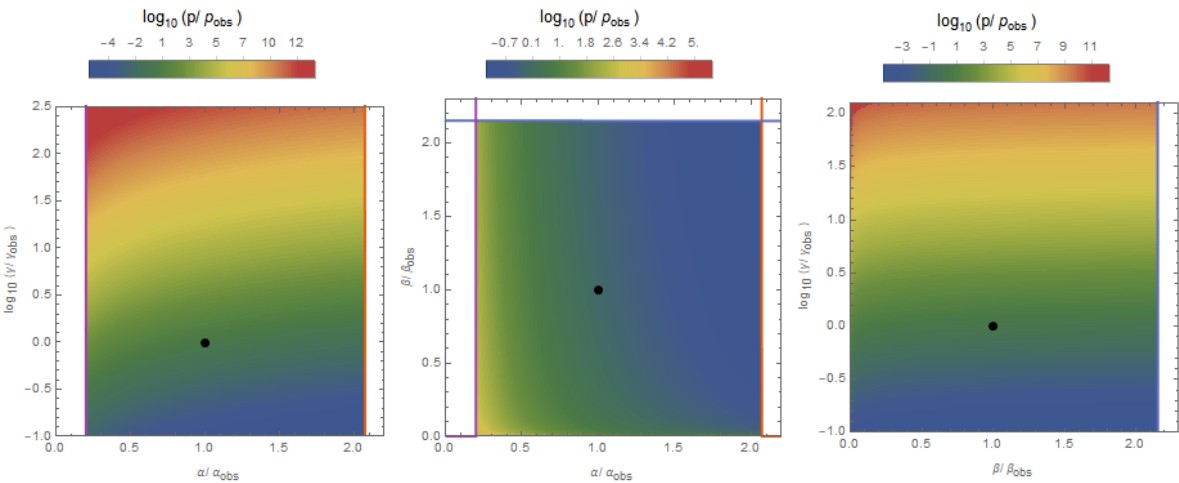

**Figure 5.** Distribution of observers with volcanic extinctions included.

## 5. Gamma Ray Bursts

### 5.1. Gamma Ray Bursts and Extinctions

Stellar explosions, in the form of supernovae, are essential for the creation and redistribution of heavy elements within our universe. However, this mechanism also gives rise to much more powerful events as well, which are orders of magnitude stronger than a typical supernova, and accompanied by an initial pulse of high energy gamma ray radiation, called a gamma ray burst (GRB). While these bursts only last several seconds to minutes in duration, they are so extreme that an event that occurs a substantial fraction of the galaxy away may be capable of throwing the ecosystem of a fragile environment such as our planet into complete disarray [51,52].

GRBs are not all identical, but instead come with a distribution of properties. Their luminosity follows a broken power law that rises for small energies, peaks at the value $L_0 = 10^{52.5}$ erg/s, and then decreases for luminosities beyond this value [53]. The initial pulse of gamma rays would only be for around ten seconds, though the spread in duration is large as well. At a deadly distance, this imparts less than an order of magnitude more than the natural fluence (time integrated energy flux) provided by the sun in visible wavelengths, so animals outside at the time would scarcely notice a major difference (if it happened during the day). However, the effects do not end with the subsidence of the initial burst.

Because most of the energy is delivered in the form of high energy photons, as they pass through the atmosphere they will ionize a large number of atoms. Each of these ions will then convert atmospheric nitrogen $N_2$ into nitrous oxide molecules $NO_x$ [54]. It is this significant buildup of 'noxious gases' that ultimately leads to effects that can last for months to years. Nitrous oxide molecules react with ozone through the reaction $NO_x + O_3 \rightarrow NO_{x+1} + O_2$, causing its depletion over a sustained period. For the fluence of $100 \, \text{kJ}/\text{m}^2$, this results in a depletion of the ozone by 50% [55]. Ozone is an excellent UV absorber, and on Earth acts as a shield against otherwise harmful solar radiation. A depletion by this amount will lead to an increase in UV flux by a factor of 3. This increased level of radiation would lead to an enhanced mutation rate among all cells exposed. This would merely lead to an enhanced cancer rate among multicellular organisms like ourselves, but would devastate microscopic organisms such as algae and plankton. Any communities exposed to such harsh radiation may collapse which, through an immense trophic cascade, could trigger the collapse of the entire ecosystem that relies on these organisms for food.

Not every ecosystem would be affected by this, however. Deep sea and benthic (ocean floor) communities are shielded from this radiation by the UV absorption of water. Any ecosystem that is ultimately reliant on the sun, however, will experience this effect, and since the sun is the largest free energy reservoir on our planet, it is naturally responsible for the majority of the complexity we observe. Even among terrestrial and surface ocean communities, atmospheric circulation simulations [54] indicate that the enhancement of ultraviolet light is confined to mid latitudes, leaving the poles relatively untouched. There they also vary the time of day, month, incident latitude, fluence of the event, and atmospheric composition. These variables have some effects on the overall atmospheric response, but none important enough to alter the qualitative conclusion of allowable flux. In [56] the duration of the burst was also shown to have no effect, as each photon acts relatively independently of the others, on a timescale that is long compared to the ionization time, and short compared to the chemical buildup time. The energy of the photons also only marginally affects the final result, since they are all very much above the ionization threshold. Thus, the quantity of primary relevance is the total fluence: only if enough photons to appreciably deplete the ozone layer are incident will drastic effects occur. The value of the fluence required to significantly perturb the ecosystem is $100 \, \text{kJ}/\text{m}^2$ [55]. For a GRB at the peak value of the luminosity distribution, this corresponds to it being situated a distance 2 kpc away.

In fact, such an event may have been responsible for one of the several mass extinction events that the Earth went through in the past half a billion years. The Ordovician mass extinction, which was the second worst in terms of genera that went extinct, seems particularly compatible with a GRB trigger [8]. Firstly, in was marked by a sudden glaciation in an otherwise climatically stable period, which may have been caused by a GRB. Of the aquatic phyla affected, those that spent more time near the surface of the water seem to have been selected against. The Ordovician extinction is also unique in that afterwards the planet was repopulated by 'high latitude survivors'. There is also some indication that the repopulation happened on land before the water communities recovered. This would be consistent with a resultant nitrate rain that would have been the outcome of a GRB, which would act as fertilizer for land plants but further suppress aquatic communities. However, the cause of the Ordovician extinction remains unproven. It will ultimately be possible to conclusively link a GRB with this event based on the complete record of environmental effects the moon keeps in its regolith [57]. However, such a test of this hypothesis remains outside the foreseeable future.

In the following, we estimate the rate of deadly GRB bursts, and how this depends on the fundamental constants.

## 5.2. GRB Rate

Data from the SWIFT survey has been used [53] to estimate the rate of gamma ray bursts, arriving at the cosmic average of $1.3^{+0.6}_{-0.7}/\text{Gpc}^3/\text{yr}$. The number density of galaxies has then been used to estimate that within the Milky Way, one GRB occurs about every $10^7$ years. This is shorter than the

typical time between mass extinctions, but most GRBs that occur even within our galaxy would be outside the sphere of influence necessary to affect life.

There are several reasons why such a simple extrapolation of the cosmic rate may be too naive, which is why this number is treated as uncertain in much of the literature. The biggest complication is the dependence of the rate on the metallicity of the environment. GRBs roughly track the star formation rate (SFR) [58], since they are the result of very massive stars with lifetimes of only several million years. However, it was noticed in [59] that the observed GRBs show a clear preference for low metallicity environments, which can be explained by noting that several of the observed GRBs at that time were associated with type 1c supernova events. These are supernovae explosions whose hydrogen and helium envelopes are absent, indicating that the star remained well mixed enough throughout its lifetime that the outer envelopes continued to participate in nuclear fusion (as well as the final bang). This can only happen if the star was rapidly rotating up until its death, and, since the presence of metals enhances the rate of angular momentum loss, GRBs can only occur in regions below a certain threshold metallicity. This is observed to be about $Z \sim 0.1 Z_\odot$.

It was then argued that since this threshold metallicity is below the lower range of typical metallicities within our Milky Way, the rate in our galaxy should be suppressed relative to the naive extrapolation from the cosmic rate. However, it was pointed out in [60] that this is incompatible with the fact that an object that looks like a GRB remnant has been detected within our galaxy that is only ten thousand years old [61]. In fact, our galaxy is continually replenished with low metallicity gas from infalling high velocity clouds coming from the outskirts of the galactic disk [62]. These low metallicity mergers, while only about 2% of our galaxy, lead to regions of greatly enhanced star formation because the collision results in regions of locally denser gas. Because of this, [63] concludes that the naive extrapolation of galactic GRB rate is likely an *underestimate* .

To proceed, we outline a simple model that takes the GRB rate to be directly proportional to the star formation rate The fraction of stars which become GRBs, $f_{\text{grb}} \sim 10^{-7}$, will be assumed independent of physical constants at this juncture, as the theoretical underpinning of this number is not on solid enough footing to track the dependence of the constants. Star formation follows the Kennicutt-Schmidt law, $\psi_{\text{sfr}} \propto \rho^{1.4}$ [64], with quite a tight correlation over a wide range of scales, as far as astrophysical observations go. A simplistic account for this scaling can be understood in terms of a model where the star formation rate is proportional to the density divided by the free-fall time of the gas, which scales as $t_{\text{ff}} \sim (G\rho)^{-1/2}$, yielding $\psi_{\text{sfr}} = \epsilon_{\text{sfr}} G^{1/2} \rho^{3/2}$, though this neglects the many intricacies accompanying the star formation process. We note that this law seems to break down below the kiloparsec scale, but since the deadly distance is larger than this breakdown, we effectively average over regions of any smaller size than this. We model the galactic disk as having uniform density and radius $r_{\text{gal}}$, and relatively thin height $h_{\text{gal}} \sim r_{\text{gal}}/10$, so that $\rho = 10 M_{\text{gal}}/(\pi r_{\text{gal}}^3)$. Then the rate of deadly GRBs can be expressed as

$$\Gamma_{\text{grb}} = \frac{f_{\text{grb}} \, \epsilon_{\text{sfr}} \, M_{\text{gal}}}{t_{\text{ff}} \, M_\star} f_{\text{vol}} \left( \frac{r_\dagger}{r_{\text{gal}}} \right) \tag{41}$$

The fraction of GRBs which are deadly to Earth, $f_{\text{vol}}$, takes into account that the Milky Way is larger than the deadly distance $r_\dagger$, and so only a fraction of GRBs will be dangerous. This distance is $r_\dagger \sim 2$ kpc, several times smaller than the radius of our galaxy, and several times larger than its height. On this scale the galaxy is effectively two dimensional, but we introduce the (slightly crude) ramp function, which holds more generically:

$$f_{\text{vol}}(x) = \begin{cases} \frac{40}{3} x^3 & x < \frac{3}{40} \\ x^2 & \frac{3}{40} < x < 1 \\ 1 & 1 < x \end{cases} \tag{42}$$

For our values, the fraction of the galaxy that can affect our planet is $(r_\dagger/r_{\text{disk}})^2 \sim 0.03$, leading to a lethal event every few 100 Myr [65]. This is compatible with the expectation that there has been one GRB driven extinction since the emergence of complex life.

Before moving on, we note several simplifications we have used in this expression: firstly, we did not take into account that the density of the galaxy is a function of the radius, leading to a potential effect that the interior parts are expected to experience a higher rate of GRBs [66]. Additionally, we do not take into account the exponential decrease in star formation with time, arising from the gas reserves becoming depleted. With this effect, even if a galaxy initially has a GRB rate high enough to disrupt its habitable systems, the rate will ultimately drop to a low enough value that any habitable stars that are still present will be capable of supporting complex life, an observation used in [67] to argue that the universe may have only recently become habitable.

Lastly, we have focused exclusively on smaller, more numerous GRBs from within our galaxy, though more powerful GRBs from nearby dwarf galaxies are potentially relevant as well. These were the focus in [68], where they calculated the extinction rate as a function of the cosmological constant $\Lambda$. Extinctions from this type increase for smaller values of $\Lambda$ because hierarchical structure formation continues for longer. This effect may actually favor larger values of $\Lambda$, as then galaxies would be more isolated, leading a decreased extinction rate.

### 5.3. What Sets $r_\dagger$?

We now calculate the distance to which a gamma ray burst can affect a planetary atmosphere. In our universe this is 2 kpc, which is set by the requirement that the number of high energy photons rivals the amount of ozone in the planet's atmosphere. We track the dependence of both these numbers on the fundamental physics parameters in turn.

Though in actuality highly complex, the physics of gamma ray bursts can be distilled down to a very simple picture, known as the fireshell model [69]: energy densities, in the form of magnetic fields, build up in the environment of a collapsing star. Because of the near total participation in these fully convective systems, this process continues until electrons are capable of being pair produced in this environment. The subsequent annihilation produces photons in the 100 keV-MeV range, which subsequently escape the system and propagate through the universe in a highly collimated beam. If the fraction of the stellar energy converted into photons through this process is denoted by $\epsilon_{\text{grb}}$, then the number will be

$$N_{\text{grb}} = \epsilon_{\text{grb}} \frac{1}{\beta \gamma^3} \tag{43}$$

The number of incident photons on a planet a distance $r$ away will then be related to this quantity through $4\pi\alpha r^2 N_{\text{hits}} = A_\oplus N_{\text{grb}}$, where $\alpha \sim 10^{-2}$ is the opening angle of the jet (which depends very weakly on the underlying physics [70]). The blast will be lethal when $N_{\text{hits}} \sim N_{\text{ozone}}$, and so this leads to a lethal distance of

$$r_\dagger = 5\, R_\oplus \sqrt{\frac{N_{\text{grb}}}{N_{\text{ozone}}}} \tag{44}$$

To proceed, we need the total number of ozone molecules in the atmosphere. Thankfully, this is not as environmentally dependent as one might at first suppose. Recall that ozone is produced when a UV photon breaks apart an $O_2$ molecule, producing two Os that can then react with an ambient $O_2$ to produce an $O_3$ (e.g., [71]). The energy needed for this first reaction is associated with photons of wavelength shorter than 242 nm [72]. Additionally, it is relatively easy for ozone to be photodissociated, which occurs with photons of wavelength less than 1100 nm. This process proceeds until sufficient ozone has built up that these photons are effectively screened from the lower atmosphere, preventing the ozone from further destruction. This makes the column density of ozone simply equal to the inverse of its cross section, $n^c = 1/\sigma \sim 7 \times 10^{18}/\text{cm}^2$. The cross section in this wavelength region is purely geometric, being far away from any enhancements that come from resonances affecting

shorter wavelengths. That this simple picture is correct is evidenced by the fact that the ozone layer is practically constant no matter what the oxygen content of the atmosphere is over almost four orders of magnitude [71], once it reaches sufficient density to saturate this criterion. From here, it is straightforward to find the total ozone in the atmosphere by multiplying by the surface area of the planet. Then, the final expression for the lethal distance is

$$r_\dagger = 6.5 \, \frac{M_{pl}^{3/2}}{\alpha \, m_e^{3/2} \, m_p} \tag{45}$$

The coefficient has been set to match the observed value in our universe. Using expressions for the galaxy mass and radius from the appendix in [2], the extinction rate becomes

$$\Gamma_{\mathrm{grb}} = 9.1 \times 10^{-14} \, \frac{Q^{3/2} \, m_p^2}{M_{pl}} \, f_{\mathrm{vol}} \left( \frac{3.6 \times 10^4 \, \kappa^{3/2}}{Q^{1/2} \, \alpha \, \beta^{3/2} \, \gamma^{1/2}} \right) \tag{46}$$

We have included the cosmological density parameter $\kappa = 10^{-16}$ and amplitude of fluctuations $Q = 1.8 \times 10^{-5}$ for completeness, but these will be held fixed in our analysis. The distribution of observers with this rate is plotted in Figure 6.

The probabilities if GRBs are the only cause of mass extinction are

$$\mathbb{P}(\alpha_{obs}) = 0.32, \quad \mathbb{P}(\beta_{obs}) = 0.23, \quad \mathbb{P}(\gamma_{obs}) = 0.04 \tag{47}$$

Of the four possible causes of extinctions we consider, this gives the lowest probability values of all.

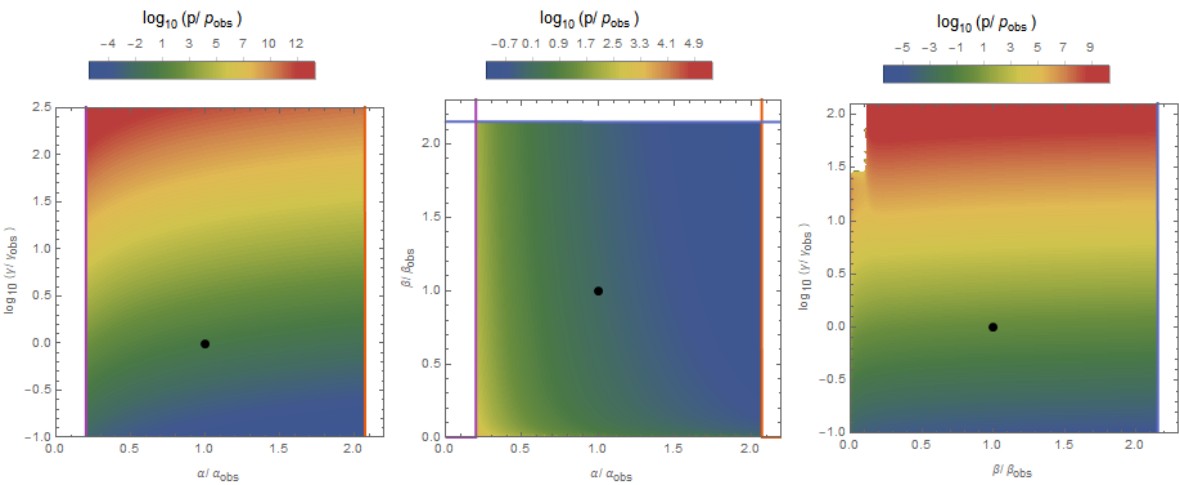

**Figure 6.** Distribution of observers with gamma ray burst-induced extinctions.

## 6. Discussion

### 6.1. Multiple Causes

Up to this point, we have derived expressions for the rates of various purported extinction causing processes, with an aim to be able to extrapolate these rates to different values of the physical constants $\alpha$, $\beta$ and $\gamma$. When we computed the probabilities of observing our values for the fiducial values $t_{\mathrm{rec}} = 10$ Myr, $\Gamma_\dagger = 1/(90 \, \mathrm{Myr})$, we found that these effects did not exert as large an influence on these quantities as the factors we discussed in our previous work, but some of the processes had a larger impact on these values than others. In particular, while glaciations and volcanoes had almost no effect on the probabilities, comets decreased $\mathbb{P}(\beta_{\mathrm{obs}})$ by a factor of 5, and GRBs decreased $\mathbb{P}(\gamma_{\mathrm{obs}})$ by a factor of 8. Here, we extend this previous analysis to include the effects of multiple extinction causes

simultaneously. Additionally, we consider the effects of using the alternative mass extinction models, and the parameterizations of the recovery timescale, discussed in Section 2.

In Table 1 we present values for the setback model, given by Equation (2). The most interesting thing to note is that including GRBs as a factor uniformly makes the probability of observing our strength of gravity below 10%. The worst case is when GRBs are taken to be the only cause of mass extinctions, and several of the combinations are only just below 10%, but it is certainly fair to say that our position in this universe is better explained without including this hazard. The prediction we can derive from this is that technosignatures should not be more prevalent in GRB-quiet regions of the galaxy. Also of note is that taking the recovery time to be set by the length of the year usually makes $\mathbb{P}(\gamma_{\mathrm{obs}})$ about 2 times lower, but $\mathbb{P}(\beta_{\mathrm{obs}})$ several times higher. The dependence of generation time on the length of the year may in principle be detectable remotely too, for example by looking at seasonal variations of biosignature gases. Since year length shows extreme variation from planet to planet, this effect could be quite noticeable. The probabilities with only comets, for instance, are increased with this choice, but the effect is too mild to make any predictions based on this.

**Table 1.** Probabilities for various extinction hypotheses discussed in the text. Here the reset model is used for the two choices of the recovery time and including all possible combinations of extinction processes. In all of the above, the recovery time is set to be 10 Myr and the total extinction rate is $(90\,\mathrm{Myr})^{-1}$.

| | **Setback Model** | | | | | |
|---|---|---|---|---|---|---|
| $t_{\mathrm{rec}}$ | **Molecular** | | | **Year** | | |
| | $\mathbb{P}(\alpha_{\mathbf{obs}})$ | $\mathbb{P}(\beta_{\mathbf{obs}})$ | $\mathbb{P}(\gamma_{\mathbf{obs}})$ | $\mathbb{P}(\alpha_{\mathbf{obs}})$ | $\mathbb{P}(\beta_{\mathbf{obs}})$ | $\mathbb{P}(\gamma_{\mathbf{obs}})$ |
| comets | 0.125 | 0.0809 | 0.287 | 0.169 | 0.217 | 0.117 |
| grbs | 0.323 | 0.233 | 0.042 | 0.241 | 0.277 | 0.019 |
| volcanoes | 0.193 | 0.426 | 0.303 | 0.192 | 0.425 | 0.293 |
| glaciations | 0.193 | 0.423 | 0.304 | 0.193 | 0.426 | 0.305 |
| comets+grbs | 0.208 | 0.0761 | 0.0757 | 0.207 | 0.242 | 0.0284 |
| comets+glaciations | 0.123 | 0.116 | 0.267 | 0.169 | 0.25 | 0.154 |
| comets+volcanoes | 0.124 | 0.115 | 0.265 | 0.169 | 0.25 | 0.148 |
| grbs+glaciations | 0.299 | 0.253 | 0.0527 | 0.233 | 0.289 | 0.0309 |
| grbs+volcanoes | 0.299 | 0.253 | 0.052 | 0.232 | 0.289 | 0.0301 |
| glaciations+volcanoes | 0.193 | 0.423 | 0.303 | 0.193 | 0.426 | 0.299 |
| comets+grbs+glac | 0.198 | 0.1 | 0.0905 | 0.207 | 0.255 | 0.037 |
| comets+grbs+vol | 0.198 | 0.0998 | 0.0895 | 0.206 | 0.256 | 0.036 |
| comets+glac+vol | 0.125 | 0.146 | 0.262 | 0.169 | 0.274 | 0.171 |
| grbs+glac+vol | 0.283 | 0.262 | 0.0697 | 0.228 | 0.297 | 0.0387 |
| all | 0.193 | 0.123 | 0.0995 | 0.205 | 0.266 | 0.043 |

Table 2 displays the same results for the reset model, from Equation (4). The probabilities are broadly similar with those of the setback model, since their functional dependences are so similar. Table 3 reports on the intermediate disturbance model, Equation (5): from here one can see that the problems that plague the other two models are largely absent in this one, the majority of the probabilities being either above 10%, or nearly so. From this, we conclude that this model of mass extinctions fares the best with the multiverse hypothesis.

**Table 2.** Probabilities for different combinations of extinction processes and recovery times with the reset model. Here the noogenesis timescale is set to be 100 Myr.

| | Reset Model | | | | | |
|---|---|---|---|---|---|---|
| $t_{noo}$ | Molecular | | | Year | | |
| | $\mathbb{P}(\alpha_{obs})$ | $\mathbb{P}(\beta_{obs})$ | $\mathbb{P}(\gamma_{obs})$ | $\mathbb{P}(\alpha_{obs})$ | $\mathbb{P}(\beta_{obs})$ | $\mathbb{P}(\gamma_{obs})$ |
| comets | 0.105 | 0.0717 | 0.251 | 0.168 | 0.26 | 0.136 |
| grbs | 0.294 | 0.243 | 0.0332 | 0.226 | 0.293 | 0.0194 |
| volcanoes | 0.19 | 0.436 | 0.317 | 0.188 | 0.432 | 0.31 |
| glaciations | 0.192 | 0.43 | 0.312 | 0.19 | 0.43 | 0.31 |
| comets+grbs | 0.179 | 0.0804 | 0.0617 | 0.202 | 0.273 | 0.0351 |
| comets+glaciations | 0.114 | 0.132 | 0.248 | 0.167 | 0.288 | 0.174 |
| comets+volcanoes | 0.114 | 0.131 | 0.247 | 0.168 | 0.289 | 0.168 |
| grbs+glaciations | 0.265 | 0.262 | 0.0506 | 0.22 | 0.308 | 0.038 |
| grbs+volcanoes | 0.265 | 0.262 | 0.0497 | 0.22 | 0.309 | 0.0368 |
| glaciations+volcanoes | 0.191 | 0.431 | 0.313 | 0.189 | 0.431 | 0.31 |
| comets+grbs+glac | 0.175 | 0.119 | 0.07 | 0.201 | 0.288 | 0.0428 |
| comets+grbs+vol | 0.175 | 0.119 | 0.0692 | 0.201 | 0.288 | 0.042 |
| comets+glac+vol | 0.12 | 0.189 | 0.257 | 0.167 | 0.303 | 0.19 |
| grbs+glac+vol | 0.25 | 0.271 | 0.0616 | 0.217 | 0.317 | 0.0467 |
| all | 0.173 | 0.145 | 0.0874 | 0.2 | 0.297 | 0.0504 |

**Table 3.** Probabilities for different combinations of extinction processes and recovery times with the intermediate disturbance hypothesis model. Here the disturbance timescale is set to be 10 Myr.

| | Intermediate Disturbance Hypothesis Model | | | | | |
|---|---|---|---|---|---|---|
| $t_{dist}$ | Molecular | | | Year | | |
| | $\mathbb{P}(\alpha_{obs})$ | $\mathbb{P}(\beta_{obs})$ | $\mathbb{P}(\gamma_{obs})$ | $\mathbb{P}(\alpha_{obs})$ | $\mathbb{P}(\beta_{obs})$ | $\mathbb{P}(\gamma_{obs})$ |
| comets | 0.106 | 0.119 | 0.251 | 0.151 | 0.462 | 0.4 |
| grbs | 0.2 | 0.329 | 0.0984 | 0.129 | 0.384 | 0.0963 |
| volcanoes | 0.128 | 0.356 | 0.368 | 0.156 | 0.404 | 0.262 |
| glaciations | 0.131 | 0.323 | 0.337 | 0.148 | 0.466 | 0.455 |
| comets+grbs | 0.165 | 0.135 | 0.0818 | 0.18 | 0.453 | 0.124 |
| comets+glaciations | 0.113 | 0.205 | 0.256 | 0.144 | 0.498 | 0.454 |
| comets+volcanoes | 0.113 | 0.202 | 0.261 | 0.147 | 0.496 | 0.489 |
| grbs+glaciations | 0.185 | 0.353 | 0.113 | 0.142 | 0.414 | 0.146 |
| grbs+volcanoes | 0.185 | 0.351 | 0.113 | 0.141 | 0.423 | 0.162 |
| glaciations+volcanoes | 0.129 | 0.345 | 0.359 | 0.153 | 0.435 | 0.4 |
| comets+grbs+glac | 0.16 | 0.18 | 0.132 | 0.175 | 0.46 | 0.158 |
| comets+grbs+vol | 0.16 | 0.179 | 0.131 | 0.179 | 0.472 | 0.166 |
| comets+glac+vol | 0.123 | 0.281 | 0.274 | 0.147 | 0.438 | 0.489 |
| grbs+glac+vol | 0.167 | 0.349 | 0.178 | 0.138 | 0.443 | 0.186 |
| all | 0.159 | 0.227 | 0.154 | 0.172 | 0.482 | 0.187 |

Additionally, we report on the effects of varying the extinction rate and recovery time: since the values of both $t_{rec}$ and $\Gamma_\dagger$ are uncertain, we report the largest value of $\Gamma_\dagger t_{rec}$ for which all values of the probabilities are greater than 10%. This is shown in Table 4. In this table, we have actually fixed $t_{rec} = 0.1 t_{noo} = t_{dist} = 10$ Myr, and report the smallest values for $1/\Gamma_\dagger$ for which all probabilities are greater than 0.1, in Myr. Since for the first two columns, taking the rate of mass extinctions to be very small recovers the scenario where they can be neglected entirely, this minimal value is guaranteed to exist. In the intermediate disturbance hypothesis case, this is not guaranteed, as this scenario favors values for which $\Gamma \sim 1/t_{dist}$, but nevertheless this value does exist for all cases we consider. Though our results are phrased in terms of a fixed recovery time and letting the rate vary, all quantities are only dependent on the product of these, so that if the value $x$ is reported in the table, the actual restriction is $\Gamma_\dagger t_{rec} < 10/x$. If one prefers to hold the extinction rate fixed at the observed value and contemplate

varying the recovery time, one has $t_{noo} > 900/x$ Myr. Though in this case $t_{noo}$ also appears in the exponent of the expression for $f_{int}$, the effects of varying this are comparatively small.

Several features can be extracted from this table. Firstly, for many combinations, the smallest allowable extinction time is less than the observed estimate of 90 Myr. For glaciations and volcanoes in particular, the extinction timescale could be 10 Myr (the smallest allowable by our formalism), and still we would be in this universe. The reason for this is that universes with lower extinction rates were bad real estate anyway, being disfavored regions of the probability distribution for other reasons. Combining multiple processes always results in roughly an average of the lone extinction times. Though we made the simplification that, when multiple effects are presented, each individual rate is taken to be the same, interpolating to more general mixtures can be made by using this observation.

**Table 4.** The smallest extinction interval $\Gamma_\dagger^{-1}$ for which all values are above 10%. All values in Myr.

| | Setback | | Reset | | IDH | |
|---|---|---|---|---|---|---|
| $t_{rec}$ | mol | Year | mol | Year | mol | Year |
| comets | 140 | 71 | 132 | 55 | 70 | 12 |
| grbs | 424 | 1061 | 422 | 813 | 96 | 95 |
| volcanoes | 10 | 10 | 10 | 10 | 10 | 10 |
| glaciations | 10 | 10 | 10 | 10 | 10 | 10 |
| comets+grbs | 160 | 640 | 201 | 473 | 102 | 68 |
| comets+glaciations | 74 | 40 | 69 | 32 | 39 | 10 |
| comets+volcanoes | 75 | 45 | 70 | 36 | 40 | 13 |
| grbs+glaciations | 216 | 537 | 215 | 412 | 46 | 58 |
| grbs+volcanoes | 218 | 540 | 217 | 415 | 50 | 52 |
| glaciations+volcanoes | 10 | 10 | 10 | 10 | 10 | 10 |
| comets+grbs+glac | 114 | 430 | 138 | 319 | 69 | 52 |
| comets+grbs+vol | 117 | 433 | 139 | 321 | 69 | 47 |
| comets+glac+vol | 53 | 33 | 49 | 27 | 29 | 10 |
| grbs+glac+vol | 148 | 364 | 148 | 280 | 33 | 41 |
| all | 91 | 328 | 107 | 243 | 53 | 41 |

One simplification we have made in our analysis is that the rates of all these processes were treated as constant throughout the universe. In fact, they all are most likely environmental: the Oort clouds of stars born in large clusters can be severely depleted [73], galaxies and regions of galaxies with lower star formation rate will have less GRBs, and planets with less internal heat will have less glaciations and volcanoes. If there truly is a selection pressure for quiet environments, this intra-universe variability will surely play a role. Our analysis has been wholly complementary to this line of reasoning; it serves to investigate how much the coincidence of these timescales can be explained by multiverse reasoning, but not necessarily how much must. Considering both these selection effects in unison is worth further investigation.

*6.2. Why Are We in This Universe?*

The answer to this question can depend on a great number of factors. In a multiverse context, the probability of being in a particular universe is directly proportional to the number of observers in that universe. The trouble is, there is a large number of things this could depend on, covering a range of scales, from the subatomic to the cosmic. In order to begin to address this question, it is necessary to get a rough idea for what the most important factors are in controlling the total number of intelligent beings in our universe. In this initial series of four papers, we have surveyed a host of different potential controlling processes, and are now in a position to gauge the relative importance of each. The discussion, representing an initial attempt to make progress on this question, has been necessarily inchoate. Very many of the factors involved have only been crudely represented in this analysis, and a great many more have been omitted altogether. Nevertheless, several key insights have been gained, and additionally, this framework can be used as a scaffolding to incorporate arbitrarily

sophisticated criteria for the creation and development of complex life. At the close of this first attempt, let us reflect on the generic lessons that have been learned.

The tools for estimating the total number of observers in the universe have been around for decades in the form of the Drake equation. Furthermore, even though we may not have a good idea on the absolute magnitude for some of the factors, often it is possible to determine how each will depend on the physical constants, given a criteria for habitability. Of these, some factors were more sensitive to the laws of physics, and to the assumptions about what life needs that were put in. To recapitulate our results: the number of habitable stars in the universe is the backbone of this computation, and this factor exerts a pressure to live in universes with stronger gravity that must be overcome by one of the other factors in order to provide a consistent picture. The fraction of stars that have planets, on the other hand, was relatively insensitive to the laws of physics, and to the assumptions about planet formation and galactic evolution that we made. Likewise, the properties of planets is likely not a key factor for determining which universe we live in. By far the most important factor was found to be the fraction of planets that develop life. This was most sensitive to the assumptions made, and led to the largest number of predictions for the distribution of life throughout our universe. The fraction of planets that develop intelligence can be similarly constraining, as found with the few different models we explored in [3]. The follow-up we performed here, detailing the purported stymieing effects of mass extinctions, does not play as large a role, but the effects can still be nontrivial.

There is one final factor which we have not discussed, which is the average number of observers per civilization. In principle this may lead to drastic changes in our conclusions, if this depends sensitively on physics. However, we refrain from incorporating this into our analysis at the present moment, largely because it is hard to say anything concrete about this factor without veering into the realm of wild speculation. One thing that may be noted is that it is a perfectly consistent prescription to neglect this factor altogether, as in [74]. The viewpoint here is that "it takes a village to raise a question": that is, that the consciousness you enjoy is not wholly your own, but is in part inherited from the whole history of society. By shifting the selection pressure onto the civilization rather than the individual, this sidesteps this complication completely. While controversial (see [75,76]), this is the de facto stance we have adopted in these papers.

While the exploration of these topics has sometimes resulted in discovering that some factors lead to more predictions than others, it was necessary to establish which of these factors were the most important early, in order to guide the direction of future research. From this, our recommendation would be to look most closely at the properties of stars and the factors that influence the origin of life, as these have generated the most predictions so far. This is not to say that the others are not expected to yield any interesting results at all: indeed, the nature of the task at hand is that any criterion, however innocuous seeming, may be found to be the absolute key driving factor for why we arose in this universe. Thankfully (for these purposes), we have a few decades ahead of us before we start to measure a robust number of exoatmospheres, so there is potentially ample time to sort out the key influences in this proposal.

It should be stressed that a substantial fraction of habitability criteria are incompatible with the multiverse. Then, if the multiverse scenario is true, this leads to the prediction that future surveys will determine these criteria are wrong. In addition, while each individual instance of prediction is a far cry from proving that other universes exist, by now we have accumulated a list of them: taken together, the ultimate case for the multiverse can be made far stronger. The more factors we incorporate, the more predictions we will be able to make, and the stronger our case will be. Though this method will never be able to do better than an indirect inference of the existence of other realms forever outside our reach, in the absence of a direct way of observing them, it represents the best path forward.

Through the course of our analysis we found the additional complication that some habitability criteria are only compatible with the multiverse if others are simultaneously employed. This was the case, for instance, when the requirement of plate tectonics was found incompatible on its own, but consistent when the tidal locking condition was included. This conditional interdependence

prevents us from testing each criterion in complete isolation, instead necessitating a check of its compatibility with all previously considered hypotheses. This leads to an exponential proliferation of different choices, which is already becoming overly cumbersome to enumerate, test and report on.

Additionally, attention so far has been restricted to just three physical constants, which do a fair job of determining the character of the macroscopic world. However, for full consistency, about 10 of the parameters of the standard models of particle physics and cosmology must be incorporated. Furthermore, more attention needs to be paid to local environmental variations within the universe, which to this point has crudely been represented only by stellar mass. Expanding the calculation in these ways will significantly increase computational costs, but promises to extend our predictive power. Of equal importance will be to identify further criteria for what life needs and how these processes likely depend on physics to incorporate into this analysis. The more criteria are put into the system, the more predictions will be returned, and the stronger the case either for or against the multiverse will be.

## 7. Conclusions

In this work, we have investigated the influence of mass extinctions on the probability of our presence in this universe, within the multiverse context. On the whole, we find that this factor is not as important as the others in the Drake equation, as explored in other papers of this series, and so we do not expect the extinction rate to be the determining factor for why we live here. However, depending on the assumptions we made about the relative importance of the various extinction mechanisms we consider and their overall effect, more can be said. Firstly, taking mass extinctions to be solely caused by gamma ray bursts led to an uncomfortably small probability for observing our strength of gravity, signaling that this assumption should be wrong. In contrast, taking comets to be the only cause had a much smaller impact on the probabilities, and the geologic influences had almost none at all. Combining various processes usually tempers the effects each would exert individually. Various models for extinction effects were explored, and while the setback and reset model were broadly similar, the intermediate disturbance hypothesis was very forgiving, resulting in probabilities almost always compatible with our existence here. Lastly, taking the recovery time to be dictated by the year length rather than the molecular timescale changed some of the probability values by as much as a factor of a few, but not drastically. So, while certain specific assumptions are incompatible with the multiverse hypothesis, for the most part we can expect the selective influence exerted by extinction events to be minimal.

**Funding:** This research received no external funding.

**Acknowledgments:** I would like to thank Fred Adams, Gary Bernstein, RJ Graham, Mario Livio, Aki Roberge, and Alex Vilenkin for useful discussions.

**Conflicts of Interest:** The author declares no conflict of interest.

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
