# Peer review of "Multiverse Predictions for Habitability: Fraction of Life That Develops Intelligence"

_universe, doi:10.3390/universe5070175_

Reviewer 1 Report

The author looks at mass extinctions and whether this might provide some information concerning the multiverse.

This is quite an interesting article. However, there are various assumptions/omissions that I feel are very detrimental.

1)The author assumes that life must start over from scratch. To say that this is debatable is putting it mildly. The current favored model is that life did not start from scratch, and there is substantial evidence for this, such as DNA studies and extremophiles.

2)The author notes "other factors" but does not discuss what they are nor what effect they may have (before Figure 1).

3)The author's github page is not working: so it is not possible to look at the code, which should be discussed in the paper.

https://github.com/mccsandora/Multiverse-Habitability-Handler

For the most part, the author discusses various mass extinction scenarios and determines that nearly all are compatible with the multiverse, assuming assumptions such as (1) which again are disfavored. The author concludes that

"Then, if the multiverse scenario is true, this leads to the prediction that future surveys will determine these criteria are wrong. And, while each individual instance of prediction is a far cry from proving that other universes exist, by now we have accumulated a list of them: taken together, the ultimate case for the multiverse can be made far stronger."

Unfortunately there is nothing very convincing in the paper one way or another. As it stands, I feel that the paper could be greatly shortened and simply state "Maybe we live in a Universe with less mass extinctions than others, if a multiverse exits, but maybe not". Also, with slight tweaking it is possible to change the probabilities in Table 1 and Table 2 .

While a very original idea, I feel that there needs to be a good amount of work done before this this can be considered scientifically plausible. I hope some of my comments might help the author.

Author Response

I thank the reviewer for their suggestions on the draft, and find that incorporating them does improve the paper.

The author looks at mass extinctions and whether this might provide some information concerning the multiverse. 

This is quite an interesting article. However, there are various assumptions/omissions that I feel are very detrimental. 

1)The author assumes that life must start over from scratch. To say that this is debatable is putting it mildly. The current favored model is that life did not start from scratch, and there is substantial evidence for this, such as DNA studies and extremophiles.

This is a misapprehension on the reviewer’s part: throughout the paper I consider three separate models of the effects mass extinctions have on the ecosphere, and the one described, which I’ve dubbed the `reset model’, represents only one of them.  Throughout most of the paper the `setback model’ is used instead, which reflects the view that mass extinctions lead to reduced biodiversity for 10 Myr before recovery.  The reset model is then only touched upon in a table in the discussion section.

I assume most of the responsibility for this confusion, and have taken pains to increase the clarity of the exposition of this point.  I’ve added discussions to this effect on line 106, 153, 324, and 513.

2)The author notes "other factors" but does not discuss what they are nor what effect they may have (before Figure 1). 

I now mention some of these other factors and their expected effects explicitly in line 126, and refer the reader to the third paper of this series, where some are considered.

3)The author's github page is not working: so it is not possible to look at the code, which should be discussed in the paper. 

https://github.com/mccsandora/Multiverse-Habitability-Handler

This link is now operational, so the code may be used and inspected at will.

For the most part, the author discusses various mass extinction scenarios and determines that nearly all are compatible with the multiverse, assuming assumptions such as (1) which again are disfavored. The author concludes that 

"Then, if the multiverse scenario is true, this leads to the prediction that future surveys will
determine these criteria are wrong. And, while each individual instance of prediction is a far cry from proving that other universes exist, by now we have accumulated a list of them: taken together, the ultimate case for the multiverse can be made far stronger." 

It is worth stressing that the final subsection, `Why are we in this universe?’, is meant to serve as the conclusions for the whole series of four papers published in this volume of the journal.  As such, many of the predictions referenced in the above quote are interspersed throughout the other papers.  I have now included a separate conclusions section specific to this paper highlighting the various points I find throughout on line 631.

Unfortunately there is nothing very convincing in the paper one way or another. As it stands, I feel that the paper could be greatly shortened and simply state "Maybe we live in a Universe with less mass extinctions than others, if a multiverse exits, but maybe not". Also, with slight tweaking it is possible to change the probabilities in Table 1 and Table 2 . 

I fully agree with the reviewer that mass extinctions do not seem to be a determining factor for selecting which universe we inhabit, as expressed on line 587, but ask they consider the value of this observation.  While this may be considered a negative result, it informs how sensitive we should expect life to be to various extinction processes, and makes contact with recent trends in the multiverse literature that focus on extinction processes from a cosmological perspective.

Unfortunately, in order to come to this conclusion, I actually had to perform the calculations in the paper, which cannot be shortened much.  I think it may be a good idea to write a brief summary of this series that includes this sentiment, and will undertake this task before too long.

The probabilities presented were found by my best estimates for how various processes scale with physical parameters, paying careful attention to match onto observed rates.  The robustness of these values can be explored somewhat by comparing the different columns of the various charts presented, as well as comparing eqns 25-27 for the case of comets.  However, I must admit that the values I find are only as good as the assumptions I make, which is why I tried to be explicit as possible about this process.

While a very original idea, I feel that there needs to be a good amount of work done before this this can be considered scientifically plausible. I hope some of my comments might help the author.

Reviewer 2 Report

 The multiverse is highly controversial. Quite a lot of cosmologists consider this is not serious science and this should not be even discussed. I disagree with this view and I am sympathetic with the fact that this scenario is studied deeper. This article is therefore very welcome. 

As far as I can judge I don’t see any obvious mistakes in the calculations presented. I will therefore restrict myself to a few general questions, based on the arguments given in the first paper of this bunch : arXiv:1901.04614

I don’t understand the logics of "if our universe is good at something, we expect that to be important for life ».

1) Couldn’t this « something » just be neutral from the point of view of life ?

2) Couldn’t it be that even if this « something » is bad for life it is still compatible with the multiverse  juste because there are many more vacua with than without this something ?

Also, isn’t the full approach highly biased - to say the least - by the fact that we only work on what is favorable or not to life as we know it (which most probably gives the feeling that our Universe is greatly tuned for life) but, as life is by definition an adaptative process, it might just be impossible to have any idea of how it would have developed or not with other laws of physics ?

About the work presenter in the considered article, I have just one question. Why isn’t the self-destruction of an « Intelligent » civilisation considered more deeply ? The only intelligent civilisation we know is about to collapse not because of a volcano but because of its own … stupidity :). 

BTW it might be welcome to also mention that « Intelligence », etc. is used in a very highly anthropocentric sense. This is ok as a first approximation but it could be worse mentioning this more clearly.

Author Response

I thank the reviewer for their suggestions on the draft, and find that incorporating them does improve the paper.

The multiverse is highly controversial. Quite a lot of cosmologists consider this is not serious science and this should not be even discussed. I disagree with this view and I am sympathetic with the fact that this scenario is studied deeper. This article is therefore very welcome. 

As far as I can judge I don’t see any obvious mistakes in the calculations presented. I will therefore restrict myself to a few general questions, based on the arguments given in the first paper of this bunch : arXiv:1901.04614

I don’t understand the logics of "if our universe is good at something, we expect that to be important for life ».

1) Couldn’t this « something » just be neutral from the point of view of life ?

The confusion here is my fault, and comes from the desire to make the logic of this paper as slogan-esque as possible in order for it to be accessible to a general audience. The discussion on this point has already been amended in the published version of the first paper.  Here is the revised definition of `good’:

“Here, by saying that ‘our universe is good at something’, we really mean that by adopting the habitability criterion in question, our presence in this universe is probable, and, equally importantly, by not adopting it, our presence in this universe is improbable.”

As the present paper demonstrates, even if adopting a particular habitability criterion alters the probabilities, the majority influence the results by less than a factor of two.  Consequently, these do not lead to strong predictions, making these criteria almost neutral.  However, as also demonstrated, a few do lead to strong predictions.

2) Couldn’t it be that even if this « something » is bad for life it is still compatible with the multiverse just because there are many more vacua with than without this something?

This is an important point: for the microphysical parameters I consider, the number of vacua are expected to scale weakly with their values.  In contrast, the number of observers per vacuum shows much stronger dependence on the parameter values.  Because of this, our presence here should be much more strongly dictated by what life needs, rather than the availability of a particular universe.  This can be contrasted to the case of cosmological parameters, where the number of universes can depend exponentially on parameter values: for these, our presence here is explained mostly by availability, rather than the requirements of life.  This is discussed in the second paragraph of the discussion in the first paper, on page 19.

Also, isn’t the full approach highly biased - to say the least - by the fact that we only work on what is favorable or not to life as we know it (which most probably gives the feeling that our Universe is greatly tuned for life) but, as life is by definition an adaptative process, it might just be impossible to have any idea of how it would have developed or not with other laws of physics ?

Indeed I am worried about this issue.  In fact, one of the main motivations of this work is to probe assumptions about the robustness of life, rather than resigning to one of the dominant dogmas that `life is infinitely robust’ or `life is infinitely finnicky’.  While I cannot yet claim to have addressed this in full, the present paper, in exploring an aspect of this question of whether extinction rate is the prime selective force for our presence here, at least indicates that life is not particularly finnicky in this regard. 

I think the best way to answer this question is to focus on universes very much like our own, as I have done, which have environmental analogues in ours that can lead to testable predictions.  Even if life is infinitely adaptable, some universes may be better than others simply because they would be able to support more life in total.  Thus, taking all stars as equally habitable leads to a contradiction in the multiverse scenario, giving insight that life should at least be somewhat picky.  Aggregating more of these arguments can allow us to place limits on what we should expect life to be sensitive to, which can ultimately be tested through the exploration of our galaxy.

I admit I am at least as curious as the reviewer whether universes which are drastically different from our own can support life, but I agree that this question seems to be intractable for the foreseeable future.

About the work presenter in the considered article, I have just one question. Why isn’t the self-destruction of an « Intelligent » civilisation considered more deeply ? The only intelligent civilisation we know is about to collapse not because of a volcano but because of its own … stupidity :). 

The answer to this one is simple: it’s because I have no idea how.  More generally, biological innovations have been implicated for many of the extinctions in Earth’s history.  It may be possible to say something generic about how often this happens to ecological networks, but this will have to be left for future work. 

An additional route which I have more concrete plans to follow up on would consider how the potential failure modes of civilization may limit the average number of observers per civilization.  As this Drake factor is not the focus of the present paper, this discussion will have to be explored separately.

BTW it might be welcome to also mention that « Intelligence », etc. is used in a very highly anthropocentric sense. This is ok as a first approximation but it could be worse mentioning this more clearly.

Agreed.  I have added a few sentences on line 33 explicitly stating this.

Round  2

Reviewer 1 Report

I think the author significantly improved the manuscript. At this point, I feel that the manuscript can be published. I think the conclusion could still be improved a bit by being more clear. However, that is my personal view and not a requirement.

Author Response

I'll choose to leave the conclusions as they are now, as I think they summarize the main results of the text as succinctly and clearly as can be managed.